# Unveiling the A-to-I mRNA editing machinery and its regulation and evolution in fungi

Chanjing Feng[1,4], Kaiyun Xin[1,4], Yanfei Du ⬢[1,4], Jingwen Zou[1,4], Xiaoxing Xing[1], Qi Xiu[1], Yijie Zhang[1], Rui Zhang[2], Weiwei Huang[2], Qinhu Wang ⬢[1], Cong Jiang[1], Xiaojie Wang ⬢[1], Zhensheng Kang[1], Jin-Rong Xu ⬢[3] & Huiquan Liu ⬢[1] ✉

A-to-I mRNA editing in animals is mediated by ADARs, but the mechanism underlying sexual stage-specific A-to-I mRNA editing in fungi remains unknown. Here, we show that the eukaryotic tRNA-specific heterodimeric deaminase FgTad2-FgTad3 is responsible for A-to-I mRNA editing in *Fusarium graminearum*. This editing capacity relies on the interaction between FgTad3 and a sexual stage-specific protein called Ame1. Although Ame1 orthologs are widely distributed in fungi, the interaction originates in Sordariomycetes. We have identified key residues responsible for the FgTad3-Ame1 interaction. The expression and activity of FgTad2-FgTad3 are regulated through alternative promoters, alternative translation initiation, and post-translational modifications. Our study demonstrates that the FgTad2-FgTad3-Ame1 complex can efficiently edit mRNA in yeasts, bacteria, and human cells, with important implications for the development of base editors in therapy and agriculture. Overall, this study uncovers mechanisms, regulation, and evolution of RNA editing in fungi, highlighting the role of protein-protein interactions in modulating deaminase function.

Adenosine-to-inosine (A-to-I) editing via deamination is a remarkable type of RNA editing as it modifies a multitude of nuclear-encoded mRNAs across the animal kingdom[1]. As inosine is recognized as guanosine (G) by the translation machinery, A-to-I editing of coding regions can result in protein recoding. A-to-I editing also occurs at positions 34 (the wobble position of the anticodon) and 37 of tRNAs. While A[37] editing has been found only in eukaryotes, A[34] editing is conserved in both eukaryotes and bacteria[2]. A-to-I mRNA editing in animals is catalyzed by enzymes of the Adenosine Deaminase Acting on RNA (ADAR) family that target specific double-stranded RNA (dsRNA) structures[1]. The ADAR family originated in the last common ancestor of extant metazoans and is a metazoan innovation[3]. All ADARs share a conserved C-terminal deaminase domain and a variable number of N-terminal dsRNA binding domains (dsRBDs) that mediate substrate recognition. The enzyme responsible for A-to-I tRNA editing is known as tRNA-specific Adenosine Deaminase (Tad) or Adenosine Deaminase Acting on tRNA (ADAT). Tad1/ADAT1 mediates A[37] editing and shares a similar adenosine deaminase domain with ADARs but lacks dsRBDs[2,4]. In bacteria, homodimeric TadA deaminizes A[34], while in eukaryotes, a heterodimer composed of the catalytic subunit Tad2/ADAT2 and the structural subunit Tad3/ADAT3 deaminizes it[2,5]. TadA, Tad2, and Tad3 all carry the catalytic motifs of cytidine/dCMP deaminases (CDAs), except that the essential glutamate (E) residue for proton shuttling in deamination is replaced by a catalytically inactive residue, usually valine (V), in Tad3[4]. Both subunits are essential for cell viability. Structural studies have shown that the C-terminal CDA domain of Tad3 interacts stably with Tad2 to form the catalytic domain of the heterodimeric enzyme, and the N-terminal Tad3 forms a tRNA binding domain that is flexibly linked to the catalytic domain[5–7].

[1]State Key Laboratory for Crop Stress Resistance and High-Efficiency Production, College of Plant Protection, Northwest A&F University, Yangling, Shaanxi 712100, China. [2]College of Life Sciences, Northwest A&F University, Yangling, Shaanxi 712100, China. [3]Department of Botany and Plant Pathology, Purdue University, West Lafayette, IN 47907, USA. [4]These authors contributed equally: Chanjing Feng, Kaiyun Xin, Yanfei Du, Jingwen Zou. ✉e-mail: liuhuiquan@nwsuaf.edu.cn

Recently, A-to-I mRNA editing has been discovered in filamentous ascomycetes, specifically during sexual reproduction[8,9]. It is a common feature in Sordariomycetes[10], one of the largest classes of Ascomycota. More than 40,000 A-to-I mRNA editing sites have been identified in the perithecia (sexual fruiting bodies) of *Neurospora crassa* and *Fusarium graminearum*, respectively[11,12]. A-to-I mRNA editing has been found to play a crucial role in fungal sexual development[11,13,14]. As fungi lack orthologs of animal ADARs and the *cis*-regulatory elements of editing are also distinct from animals[12], a different A-to-I editing mechanism must exist in fungi.

Here, we demonstrate that the eukaryotic tRNA-specific heterodimeric deaminase FgTad2-FgTad3 is responsible for transcriptome-wide A-to-I mRNA editing in *F. graminearum*, with its capacity relying on the interaction between FgTad3 and a sexual stage-specific protein named Ame1 (activator of mRNA editing). This interaction has emerged in Sordariomycetes. The key residues responsible for the interaction between FgTad3 and Ame1, as well as the innovation of mRNA editing are identified. Moreover, the FgTad2-FgTad3-Ame1 complex efficiently edits target mRNAs not only in yeasts and bacteria but also in human cell lines, advancing the development of therapeutic and agricultural tools. Our study uncovers the mechanisms, regulation, and evolution of RNA editing in fungi, emphasizing the role of protein-protein interactions in controlling deaminase function.

## Results

### FgTad2 and FgTad3 form a heterodimer

Both FgTad2 and FgTad3 contain the CDA domain with conserved residues required for catalytic activities, except that the E residue essential for catalysis was replaced by an inactive V residue in FgTad3 (Fig. 1a). Compared to orthologs from yeast, plants, and animals, the sequence upstream from the CDA domain of FgTad2 is unusually long. Besides the CDA domain (FgTad3$^{CDA}$), FgTad3 also contains an N-terminal domain (FgTad3$^{N}$) known for tRNA binding. During sexual development from 1- to 8-days post-fertilization (dpf), *FgTAD2* and *FgTAD3* showed similar expression trends (Fig. 1b). Both yeast two-hybrid (Y2H) and co-immunoprecipitation (co-IP) assays confirmed their interaction (Fig. 1c, d). FgTad2 was found to interact with the C-terminal region (203-428 aa) of FgTad3, not with the N-terminal region (1-202 aa). Therefore, FgTad2 and FgTad3 act as a heterodimer for adenosine deamination in *F. graminearum*, akin to their yeast counterparts.

### FgTad2-FgTad3 preferentially binds mRNA from highly edited genes

The expression levels of *FgTAD2* and *FgTAD3* correlated with A-to-I mRNA editing activity during sexual development (Fig. 1b). In addition to being located within a similar stem-loop structure, WebLogo analysis of all the identified A-to-I mRNA editing sites[12] and A$^{34}$ sites of inosine-modified tRNAs in *F. graminearum* showed similar base preferences (Fig. 1e). To assess the mRNA-binding capability of the FgTad2-FgTad3 complex, transformants expressing *FgTAD2*−3×FLAG were used for RNA immunoprecipitation sequencing (RIP-seq) on 7-dpf perithecia with an anti-FLAG antibody. Genes expressed during sexual reproduction were categorized into edited and unedited groups based on known A-to-I mRNA editing sites[12]. Edited genes were further divided into high-edited and low-edited groups. The footprint read density was notably higher in edited genes compared to unedited genes (Fig. 1f). Additionally, high-edited genes exhibited significantly elevated footprint read densities compared to low-edited genes. These results indicate that FgTad2-FgTad3 binds to mRNA of edited genes, with a preference for highly edited genes, suggesting their involvement in mRNA editing in *F. graminearum*.

### *FgTAD2* or *FgTAD3* overexpression boosts mRNA editing

To validate the role of FgTad2-FgTad3 in mRNA editing, over-expression of *FgTAD2* or *FgTAD3* was conducted by inserting an additional copy at a specific locus controlled by the RP27 promoter. Real-time quantitative Reverse-Transcription PCR (qRT-PCR) analysis showed that the expression levels of *FgTAD2* and *FgTAD3* in the resulting *FgTAD2$^{OE}$* and *FgTAD3$^{OE}$* transformants were increased over 20-fold in 24-h hyphae and 3-fold in 7-dpf perithecia (Fig. 1g). The lower level of overexpression in perithecia was likely attributed to the higher gene expression relative to that in hyphae (Supplementary Fig. 1). Both transformants exhibited normal growth and colony morphology. Although the perithecia produced by these transformants appeared transparent, they were still able to generate regular asci and ascospores internally (Fig. 1h). Strand-specific RNA-seq analysis indicated an increase in the number of A-to-I mRNA editing sites and their median editing levels in 7-dpf perithecia of both transformants (Fig. 1i). These results indicate that elevating the expression of either *FgTAD2* or *FgTAD3* can enhance mRNA editing during sexual reproduction.

### Both *FgTAD2* and *FgTAD3* express a sexual stage-specific S-transcript

Both *FgTAD2* and *FgTAD3* expressed two major transcript isoforms via alternative transcriptional initiation (Fig. 2a). The long (L) transcript was constitutively expressed, while the short (S) transcript was expressed specifically during the sexual stage. For both genes, the abundance of the L-transcript remained relatively constant during sexual development, while the S-transcript was continuously upregulated after 4-dpf and became the dominant transcript in the late stage of sexual development (Fig. 2b). These data suggest that the S-transcript may be the main mediator for mRNA editing.

For *FgTAD2*, the transcriptional initiation site of the S-transcript was located within the open reading frame (ORF) region of the L-transcript according to the RNA-seq data (Fig. 2a). We used 5′RACE (Rapid Amplification of 5′-Ends cDNA) to determine the precise transcriptional initiation sites of the two *FgTAD2* transcript isoforms. A PCR band with a size of ~340 bp associated with the L-transcript initiation was detected in both hyphae and perithecia, while a band with a size of ~360 bp associated with the S-transcript initiation was detected only in perithecia (Fig. 2c), confirming the unique transcription of the S-transcript during sexual reproduction. Sequencing of these PCR products revealed that the transcriptional initiation sites of the L-transcript were located between −743 and −713 bp upstream of the start codon, and those of the S-transcript were located between 280 and 310 bp downstream of the start codon of the L-transcript.

### The role of S-transcript in sexual development and mRNA editing

To verify the role of the *FgTAD2* S-transcript in sexual development and mRNA editing, we generated mutants with multiple synonymous mutations introduced in its putative promoter region (85−264 bp from the L-transcript start codon) at the native locus. This aimed to suppress S-transcript transcription without affecting the L-transcript protein. The resulting T2S$^{sil-P}$ mutant exhibited normal growth but produced smaller perithecia lacking asci and ascospores (Fig. 2d). The L-transcript expression in the T2S$^{sil-P}$ mutant exhibited no discernible changes compared to the wild type (Supplementary Fig. 2), indicating that the synonymous mutations had minimal effects on the stability of the L-transcript. These results demonstrate the critical role of the *FgTAD2* S-transcript in sexual development. When used as the female strain in crosses with the H1-GFP labeled self-sterile deletion mutant of *MAT1-1-1*[15], the T2S$^{sil-P}$ mutant produced morphologically normal asci and ascospores, displaying the expected 1:1 segregation of 8 ascospores with and without GFP signals within an ascus. These results suggest that the *FgTAD2* S-transcript functions primarily in fertile tissues, and its suppression does not notably impact female fertility.

While over 18,000 sites were identified in the wild type, we detected an average of 266 A-to-I mRNA editing sites in 7-dpf perithecia of the T2S$^{sil-P}$ mutant through strand-specific RNA-seq

analysis. Since mRNA editing predominantly occurs in asci[11], mutants lacking asci may exhibit reduced mRNA editing. To further explore the roles of the S-transcript in mRNA editing, we used deletion mutants of the two mating type genes *MAT1-1-1* and *MAT1-2-1* as controls, which demonstrate comparable defects in sexual development to the T2S[sil-P] mutant[15]. In comparison to the *mat* mutants, the T2S[sil-P] mutant showed a decrease in the number of A-to-I mRNA

editing sites (Fig. 2e). Furthermore, its median editing levels were significantly lower (Fig. 2e; Supplementary Fig. 3). These results highlight the role of the *FgTAD2* S-transcript in mRNA editing. However, it is crucial to note that since the S-transcript is primarily functional in fertile tissues, comparing it with the *mat* mutants only demonstrates the direct impact of the S-transcript on RNA editing, without revealing its complete effect.

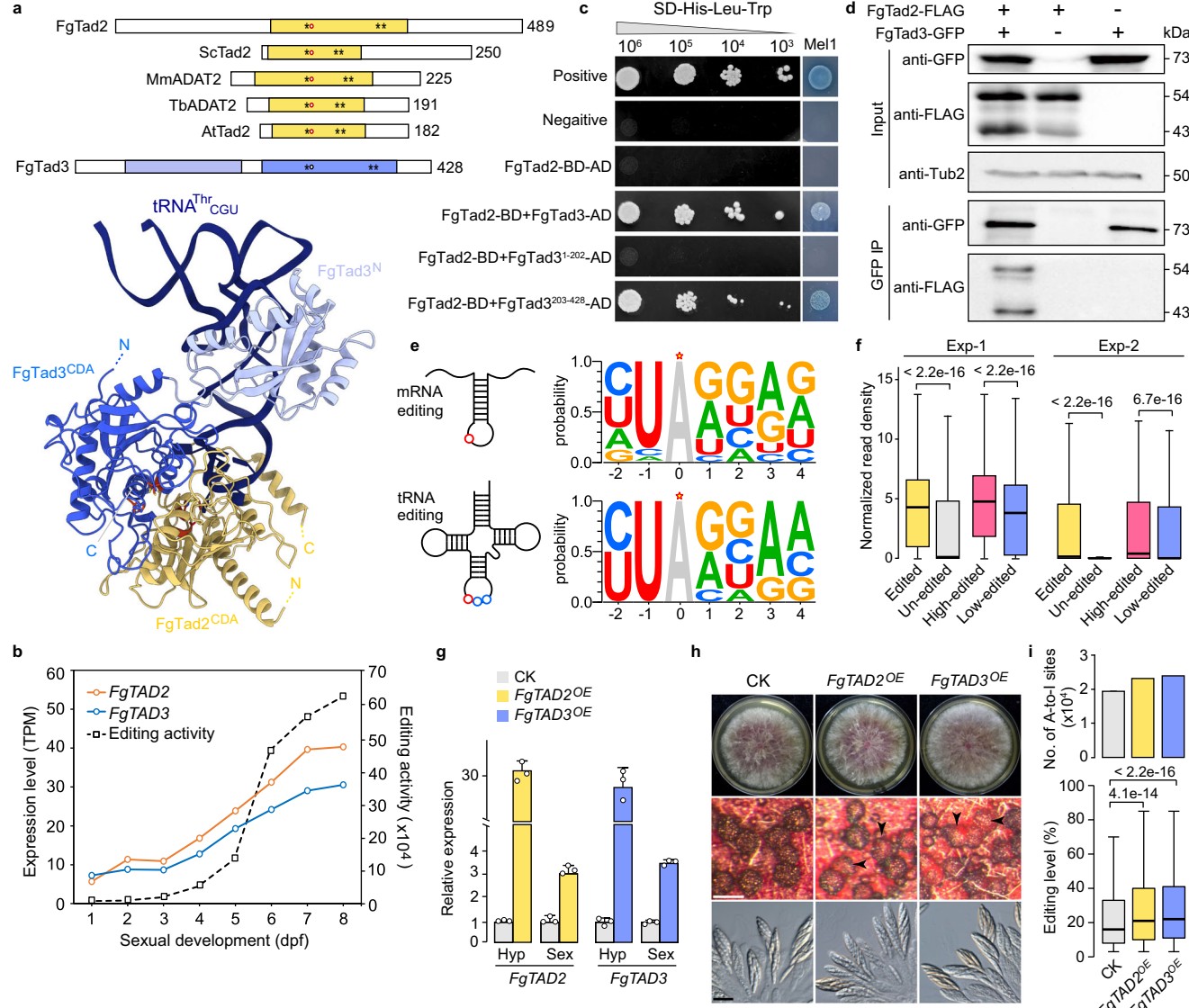

**Fig. 1 | A-to-I editing of tRNA and mRNA by the FgTad2-FgTad3 complex.**
**a** Domain structures of FgTad2 and FgTad3, along with the AlphaFold model depicting FgTad2-FgTad3 bound to tRNA. Orange, Tad2 CDA; blue, FgTad3[CDA]; sky blue, FgTad3 N-terminal domain (FgTad3[N]); Asterisk (*), Zn[2+] coordinating residue; red circle, active-site glutamate; black circle, pseudo-active-site valine. For comparison, domain structures of Tad2/ADAT2 from *Saccharomyces cerevisiae* (Sc), *Mus musculus* (Mm), *Trypanosoma brucei* (Tb), and *Arabidopsis thaliana* (At) are shown. **b** Expression levels of *FgTAD2* and *FgTAD3*, as well as editing activity (summing the editing levels of all editing sites) assessed by RNA-seq. **c** Y2H assays for the interaction between FgTad2 (BD-bait) and the full-length protein, N-terminal (1-202 aa), or C-terminal (203-428 aa) region of FgTad3 (AD-prey). Photos show growth on SD-His-Leu-Trp plates at indicated concentrations, accompanied by Mel1 α-galactosidase activity. **d** Co-IP assays for the interaction between FgTad2 and FgTad3. Total proteins (input) isolated from transformants expressing FgTad2-3×FLAG and/or FgTad3-GFP and proteins eluted from anti-GFP affinity beads (GFP IP) were detected with anti-FLAG and anti-GFP antibodies. **e** WebLogo

showing base preferences for tRNA and mRNA editing in *F. graminearum*. Editing sites are indicated by red circles or stars. **f** Boxplots of normalized read densities for edited (*n* = 6795), un-edited (*n* = 4641), high-edited (*n* = 3371), and low-edited (*n* = 3390) genes from two independent experiments (Exp-1 and Exp-2) of FgTad2-FLAG RIP-Seq. **g** Expression levels of *FgTAD2* and *FgTAD3* in hyphae (Hyp) and perithecia (Sex) of the *FG1G36140* locus deletion strain (CK) and overexpressing transformants (OE) assayed by qRT-PCR. Data are presented as mean ± SD, *n* = 3 independent replicates. **h** Morphology of colonies, perithecia, and asci/ascospores of marked strains. Arrowheads indicate transparent perithecia. White bar = 0.2 mm; black bar = 20 μm. **i** Number of A-to-I editing sites and editing levels of shared sites (*n* = 22,356) in 7-dpf perithecia of marked strains. For the bar graph, data are presented as mean of two independent replicates. For (**f, i**), boxplots indicate median (middle line), 25th, 75th percentile (box), and 1.58x interquartile range (whisker). *P* values are from two-tailed Wilcoxon rank sum tests. Source data are provided as a Source Data file.

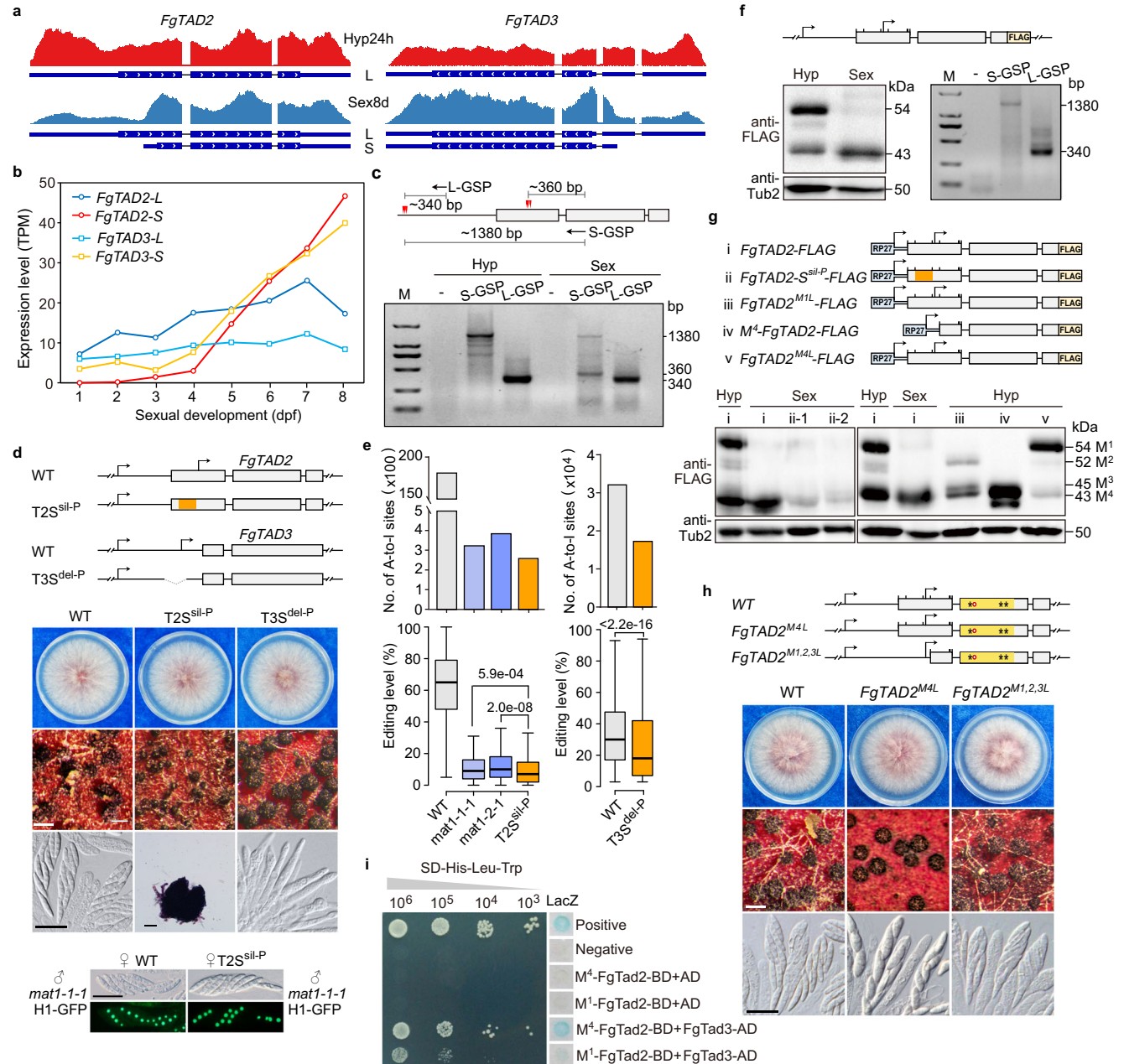

**Fig. 2 | Transcriptional and translational regulation of FgTad2-FgTad3. a** RNA-seq read coverage showing expressions of long (L) and short (S) transcripts of *FgTAD2* and *FgTAD3* in hyphae (Hyp24h) and perithecia (Sex8d). **b** RNA-seq expression levels of the L-/S-transcripts of *FgTAD2* and *FgTAD3*. **c** 5'RACE determining the transcriptional initiation sites (marked by red triangles) of the *FgTAD2* L-/S-transcripts. Expected sizes for each band are indicated. M, DNA markers; GSP, gene-specific primers. **d** Morphology of colonies, perithecia, and asci/ascospores of PH-1 (WT), T2S^sil-P^, and T3S^del-P^ mutants. Four of 8 ascospores display GFP signals in an ascus from outcrosses of marked strains. White bar = 0.2 mm; black bar = 20 μm. In schematic diagrams, arrows indicate transcriptional initiation sites, and boxes indicate CDS regions. Broken lines represent deletion regions, while the region introduced multiple synonymous mutations is highlighted in orange. **e** Number of A-to-I editing sites and editing levels of shared sites (*n* = 392 and 19,069 for left and right panels, respectively) in 7-dpf perithecia of marked strains. data are presented in bar graphs as mean of two

independent replicates. Boxplots indicate median (middle line), 25th, 75th percentile (box), and 1.58x interquartile range (whisker). *P*-values are from two-tailed Wilcoxon rank sum tests. **f** Western blots of total proteins isolated from 24 h hyphae (Hyp) and 7-dpf perithecia (Sex) of the transformant in situ expressing FgTad2-3×FLAG, probed with anti-FLAG antibodies. Only the *FgTAD2* L-transcript was detected by 5'RACE in the hyphae of this transformant. **g** Western blots of total proteins isolated from 24 h hyphae (Hyp) and 7-dpf perithecia (Sex) of transformants ectopically expressing indicated constructs. The first seven in-frame AUG (M) codons are marked with a bar on the CDS box in schematic diagrams. **h** Morphology of colonies, perithecia, and asci/ascospores of marked strains. White bar = 0.2 mm; black bar = 20 μm. The CDA domain region is highlighted in orange in schematic diagrams. **i** Y2H assays for the interaction of the FgTad2 M^1^- or M^4^-protein (BD-bait) with FgTad3 (AD-prey). Photos show growth on SD-His-Leu-Trp plates at indicated concentrations, along with the LacZ β-galactosidase activity. Source data are provided as a Source Data file.

For the *FgTAD3* S-transcript, we deleted its promoter region (−13 to −434 bp relative to the start codon) in situ. The resulting T3S^del-P^ mutant exhibited defects in ascospore formation (Fig. 2d). While the expression of the *FgTAD3* L-transcript in the T3S^del-P^ mutant showed no discernible changes compared to the wild type, the S-transcript was no longer detectable in perithecia (Supplementary Fig. 4). Both the number and median editing levels of A-to-I mRNA editing sites were markedly decreased in the T3S^del-P^ mutant compared to the wild type (Fig. 2e), indicating the importance of the *FgTAD3* S-transcript in mRNA editing. Given the essential role of both FgTad2 and FgTad3 in

RNA editing, the relatively minor impact of the absence of the *FgTAD3* S-transcript suggests that the L-transcript of *FgTAD3*, rather than *FgTAD2*, may also play important functions in fertile tissues.

### *FgTAD2* S-transcript start codon aligns with M⁴ codon of L-transcript

To assay the protein products of *FgTAD2*, we generated transformants expressing an *FgTAD2*-FLAG fusion protein in situ and detected only one protein product (~43 kDa) in 7-dpf perithecia by western blotting (Fig. 2f). Likewise, only one protein product was detected in the transformants ectopically expressing an *FgTAD2*-FLAG construct under the control of the RP27 promoter (Fig. 2g). To determine which transcript the detected protein was expressed from, we generated transformants ectopically expressing an allele with multiple synonymous mutations at the promoter region of the S-transcript. The abundance of detected proteins in the *FgTAD2*-S$^{sil-P}$-FLAG transformant was obviously decreased in perithecia compared with the transformant expressing the wild-type allele *FgTAD2*-FLAG (Fig. 2g). As the synonymous mutations did not visibly affect the L-transcript expression (Supplementary Fig. 2), these observations suggest that the detected protein in perithecia primarily originates from the S-transcript.

The transcriptional initiation sites of the *FgTAD2* S-transcript were located before the fourth in-frame AUG codon (M⁴) of the L-transcript, suggesting that the start codon of the S-transcript may align with the M⁴ codon of the L-transcript. To confirm this, we generated transformants ectopically expressing an M⁴-initiated *FgTAD2*-FLAG construct. The dominant protein product expressed by the M⁴-*FgTAD2*-FLAG transformant had the same size as the proteins detected in previous transformants (Fig. 2g), affirming that the *FgTAD2* S-transcript encodes a protein with a start codon corresponding to the M⁴ codon of the L-transcript. To investigate its functions, we introduced an A-to-C mutation at the M⁴ codon in situ. The resulting *FgTAD2^{M4L}* mutant exhibited normal growth but showed defects in ascospore formation (Fig. 2h). It produced fewer ascospores with abnormalities in asci. In contrast, mutations at the M³, M⁶, or M⁷ codon did not lead to any noticeable phenotypic changes (Supplementary Fig. 5). Therefore, M⁴ is important for the function of the *FgTAD2* S-transcript. The defect resulting from M⁴ mutations is likely due to the reduced translation efficiency of the S-transcript when using the downstream M codon as the start codon.

### *FgTAD2* L-transcript expresses two protein isoforms in hyphae

Although only the L-transcript is expressed in hyphae, two protein products (~43 and ~54 kDa) were detected (Fig. 2f, g). 5′RACE confirmed that only the L-transcript was expressed in hyphae of the in situ *FgTAD2*-FLAG transformant. Given the protein size, the two protein products were most likely expressed initiated from the M¹ and M⁴ codons, respectively. We then generated transformants ectopically expressing the *FgTAD2^{M1L}*-FLAG and *FgTAD2^{M4L}*-FLAG alleles by introducing an A-to-C mutation to the M¹ and M⁴ codons, respectively. In the *FgTAD2^{M1L}*-FLAG transformant, the M¹-protein band completely disappeared, accompanied by occurrence of two bands potentially corresponding to M²- and M³-proteins (Fig. 2g). Only trace amounts of M⁴-proteins were detected in the *FgTAD2^{M4L}*-FLAG transformant. These results suggest that the L-transcript expresses two protein isoforms by alternative translational initiation using the M¹ and M⁴ codons as start codons, respectively.

M¹-FgTad2 had a 112 aa N-terminal extension relative to M⁴-FgTad2. Both protein isoforms contained the entire CDA domain. To investigate the role of this N-terminal extension, we changed the first three M (ATG) codons into L (CTT) codons in situ simultaneously (Fig. 2h). The resulting *FgTAD2^{M1,2,3L}* mutant expected to express only the M⁴-protein had no obvious phenotypic changes, indicating that the N-terminal extension of the M¹-FgTad2 is dispensable for vegetative growth and sexual reproduction. However, Y2H assays revealed that M⁴-FgTad2

had a stronger interaction with FgTad3 compared with the M¹-FgTad2 (Fig. 2i). Therefore, the interaction of the FgTad2-FgTad3 complex is negatively affected by the N-terminal extension of the M¹-FgTad2.

### Sexual stage-specific modifications important for FgTad3 function

The two transcript isoforms of *FgTAD3* share the same ORF, with only the length of the 5′-untranslated regions (UTR) differing. Surprisingly, when an *FgTAD3*-GFP fusion protein was expressed under the control of the RP27 promoter, two smaller bands were observed in perithecia by western blotting, instead of the expected 73 kDa band detected in hyphae (Fig. 3a). The size of these bands suggests that they are unlikely to be a result of the use of alternative in-frame AUG (M) codons, as seen in *FgTAD2*. To explore the possibility of post-translational modifications on FgTad3, we performed tandem mass spectrometry (MS/MS) analysis. This analysis revealed one acetylated site (acK¹⁹⁸), two non-canonical phosphorylated sites (pE⁸ and pE²⁴¹), and one methylated site (meE¹⁷) in FgTad3 that are uniquely found in perithecia when compared to hyphae (Fig. 3b; Supplementary Fig. 6).

To verify the roles of these modifications, we generated FgTad3-hypomodified mutants by replacing each modified residue with either alanine (A) or arginine (R) to prevent potential modifications. While the *FgTAD3^{E8A}* and *FgTAD3^{E17A}* mutants did not show any observable phenotypes (Supplementary Fig. 6), the *FgTAD3^{K198R}* and *FgTAD3^{E241A}* mutants exhibited normal growth but were impaired in ascospore formation (Fig. 3c). Most of the asci in perithecia produced by these two mutants did not contain any ascospores at 7-dpf. The E²⁴¹ phosphorylation site is located within the CDA domain, while the K¹⁹⁸ acetylation site is situated in the linker region connecting the FgTad3^{CDA} and FgTad3^N domains (Fig. 3d). These results suggest that the acK¹⁹⁸ and pE²⁴¹ modifications are important for the function of FgTad3.

Additionally, we generated *FgTAD3^{K198Q}* and *FgTAD3^{E241D}* mutants by replacing K¹⁹⁸ and E²⁴¹ with glutamine (Q) and aspartic acid (D), respectively, to mimic the acetylated and phosphorylated state of FgTad3. Both mutants exhibited a slight growth defect and formed a few perithecia on carrot agar (Fig. 3c). Only a small number of elongated asci were observed in the perithecia of these mutants, with no ascospores present. These results suggest that constitutive K¹⁹⁸ acetylation and E²⁴¹ phosphorylation harm growth and perithecium formation, possibly due to the sexual stage-specific nature of these modifications.

### Identifying the activator of A-to-I mRNA editing

Since sexual stage-specific A-to-I mRNA editing occurs in both *F. graminearum* and *N. crassa*, it is expected that the potential activator of mRNA editing is unique to the sexual stage and shared between the two fungi. Using published RNA-seq data[11,16], we identified 34 orthologs expressed specifically during sexual development in both fungi (Supplementary Data 1), named sexual stage-specific conserved (SSC) genes. Eight genes with published phenotypes were not associated with mRNA editing. For the remaining 26 SSC genes, deletion mutants were generated. Deletion mutants of 6 SSC genes displayed defects in sexual development (Supplementary Data 1). We focused on *ssc23* (FG4G02710) and *ssc20* (FG3G23790) mutants with earlier stage defects. The *ssc23* mutant failed to produce perithecia, while the *ssc20* mutant formed smaller perithecia without ascogenous hyphae (Fig. 4a–c). Complemented strains had normal phenotypes. RNA-seq analysis of 60-h sexual cultures revealed an average of 152 A-to-I mRNA editing sites detected in the wild type and 26 in the *ssc23* mutant (Supplementary Table 1). Despite milder defects, no reliable A-to-I mRNA editing sites were identified in the *ssc20* mutant. RNA-seq analysis of 6-dpf perithecia of the *ssc20* mutant verified the lack of A-to-I mRNA editing (Supplementary Table 1; Supplementary Fig. 7). These results indicate the essential role of *SSC20* in A-to-I mRNA editing, leading to its designation as *AME1* (activator of mRNA editing).

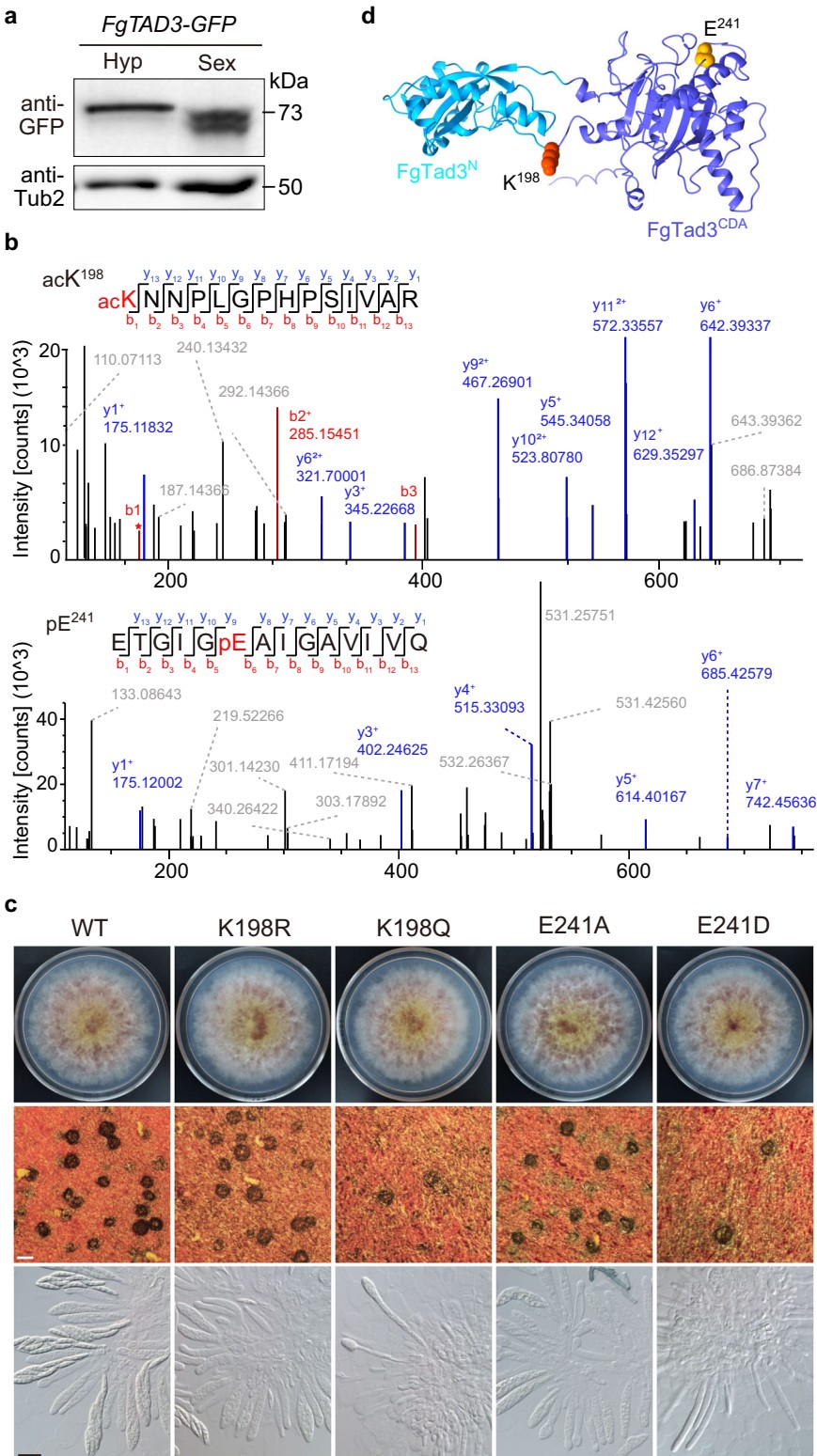

**Fig. 3 | Post-translational modifications of FgTad3. a** Western blots of total proteins isolated from 24 h hyphae (Hyp) and 7-dpf perithecia (Sex) of the transformant ectopically expressing FgTad3-GFP, probed with anti-GFP antibodies. **b** MS/MS spectrum of acetylated and phosphorylated peptides. acK$^{198}$ and pE$^{241}$ indicate the location of acetylation and phosphorylation, respectively. **c** Morphology of colonies, perithecia, and asci/ascospores of the modification-mimetic and modification-deficient mutants for the acK$^{198}$ and pE$^{241}$ of *FgTAD3*. White bar = 0.2 mm; black bar = 20 μm. **d** Cartoon model depicting the location of K$^{198}$ and E$^{241}$ sites in FgTad3.

## *AME1* expression in hyphae induces extensive A-to-I mRNA editing

During sexual development, the expression of *AME1* was continuously upregulated from 1 dpf to 5 dpf, following which the expression level tended to stabilize (Fig. 4d), aligning with the high editing activity observed after 5 dpf (Fig. 1b). To further confirm the role of *AME1* in mRNA editing, we replaced its native promoter with the RP27 promoter in situ. The resulting *AME1*-oe transformants

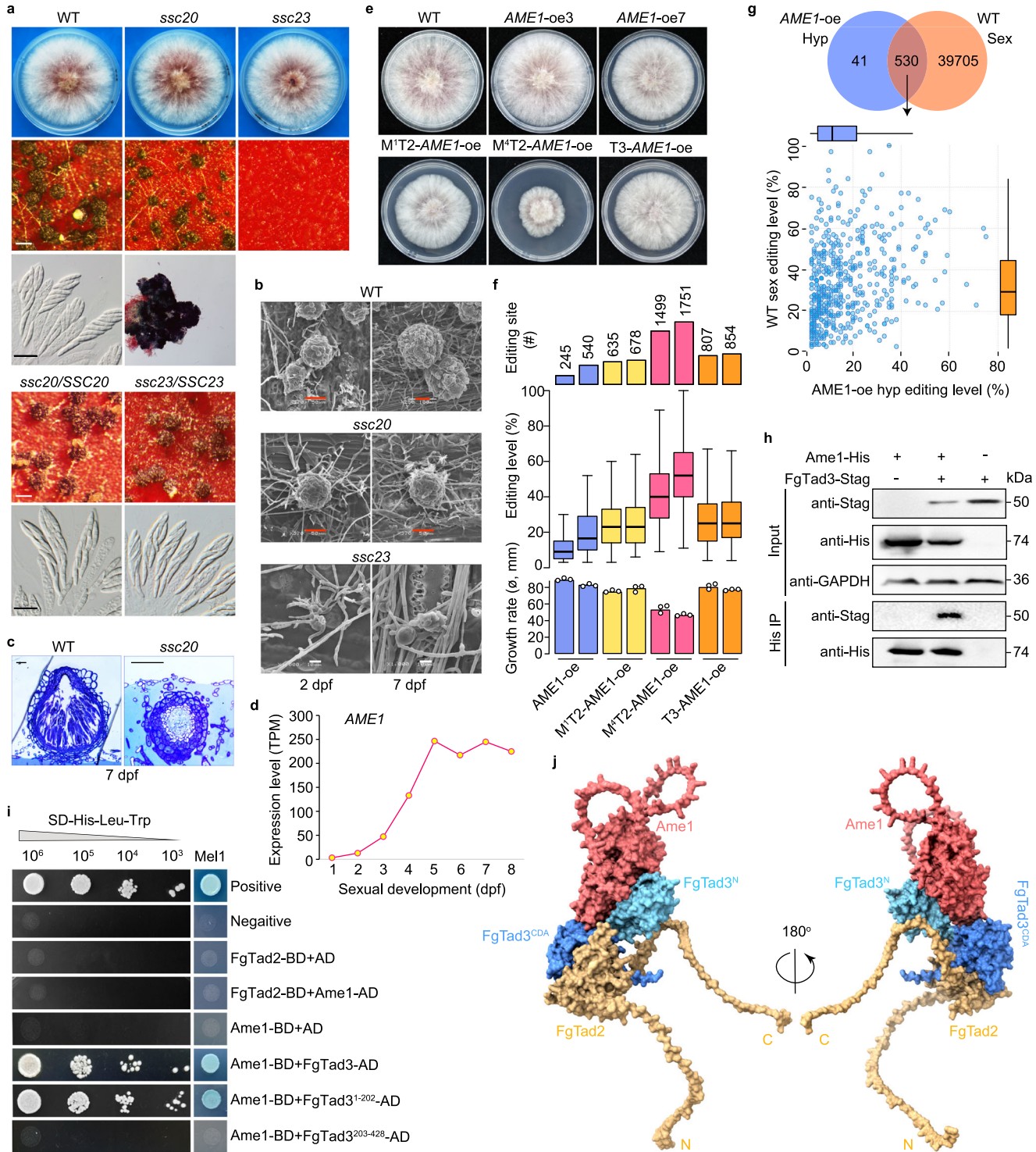

**Fig. 4 | The sexual stage-specific Ame1 essential for sexual reproduction and A-to-I mRNA editing. a** Morphology of colonies, perithecia, and asci/ascospores of deletion mutants and complemented strains of *AME1/SSC20* and *SSC23*. White bar = 0.2 mm; black bar = 20 μm. **b** Scanning electron microscope of 2-dpf and 7-dpf perithecia on wheat straw. Red bar = 50 μm; white bar = 10 μm. **c** Semi-thin sections of 7-dpf perithecia on carrot agar. bar = 20 μm. **d** RNA-seq expression levels of *AME1/SSC20*. **e** Colony morphology of transformants expressing *AME1* (*AME1*-oe3, and -oe7), or co-expressing *AME1* with M¹-*FgTAD2* or M⁴-*FgTAD2* (M¹T2-*AME1*-oe or M⁴T2-*AME1*-oe), or with *FgTAD3* (T3-*AME1*-oe) under the control of the RP27 promoter. **f** Number of A-to-I mRNA editing sites detected in 24 h hyphae, editing levels of shared sites (*n* = 182), and colony growth rates of the marked strains, with two independent transformants shown for each strain. Data of growth rates are

presented as mean ± SD, *n* = 3 independent replicates. **g** Comparison of editing levels of shared editing sites (*n* = 530) between hyphae of the *AME1*-oe transformant and perithecia of the wild type. **h** Co-IP assays for the interaction between Ame1 and FgTad3. Western blots of total proteins isolated from *E. coli* transformants expressing Ame1-6×His and/or FgTad3-Stag (input) and proteins eluted from anti-His affinity beads (His IP) were detected with anti-His and anti-Stag antibodies. **i** Y2H assays for the interaction of Ame1 (BD-bait) with FgTad2 or FgTad3, as well as FgTad3 variants (AD-prey). **j** Surface model of the ternary complex M⁴-FgTad2-FgTad3-Ame1 predicted with AlphaFold-Multimer. For (**f**, **g**), boxplots indicate median (middle line), 25th, 75th percentile (box), and 1.58x interquartile range (whisker). Source data are provided as a Source Data file.

exhibited variations in colony growth phenotypes, with some displaying wild-type characteristics while others showed slightly reduced growth rates (Fig. 4e). Interestingly, we identified numerous A-to-I mRNA editing sites in the hyphae of both types of transformants, with 245 sites in a normal transformant and 540 sites in a defective transformant (Fig. 4f), underscoring the active involvement of Ame1 in mRNA editing.

Over 90% of the editing sites detected in hyphae were also naturally edited in perithecia (Fig. 4g). These shared sites exhibited high levels of editing in perithecia but lower editing levels in hyphae. Subsequently, we overexpressed *FgTAD3* or the M$^1$/M$^4$-initiated *FgTAD2* using the RP27 promoter at a specific locus in the *AME1*-oe transformant. All resulting transformants exhibited growth defects, especially the M$^4$T2-*AME1*-oe transformants, which experienced approximately a 50% reduction in growth rates (Fig. 4e). We identified 635 to 1751 A-to-I mRNA editing sites in these transformants (Fig. 4f; Supplementary Fig. 8). The number of detected editing sites in hyphae remains much lower than that in perithecia. The number and median editing level of editing sites detected in different transformants were linked to the severity of their growth defects, indicating that mRNA editing during vegetative growth is deleterious. Furthermore, the results suggest that M$^4$-FgTad2 exhibits higher editing activity, consistent with its stronger interaction with FgTad3.

## Ame1 interacts with the N-terminal domain of FgTad3

To investigate the relationship between Ame1 and FgTad2-FgTad3, we generated transformants ectopically expressing *FgTAD3*-FLAG, *AME1*-GFP, or both constructs and performed co-IP assays. While we successfully detected FgTad3-FLAG and Ame1-GFP proteins in transformants where they were individually expressed, the detection of Ame1-GFP in the co-expressed transformant was minimal (Supplementary Fig. 9). This indicates that the simultaneous expression of both proteins is detrimental to growth and therefore repressed. We then introduced the *FgTAD3*-STAG, *AME1*-6×HIS, or both constructs into *Escherichia coli* and confirmed that Ame1 was associated with the FgTad2-FgTad3 complex in vivo through co-IP (Fig. 4h). Based on Y2H assays, it was observed that Ame1 directly interacted with FgTad3 but not with FgTad2 (Fig. 4i). Furthermore, Ame1 directly interacted with the N-terminal domain (1-202 aa) of FgTad3, while no interaction was observed with the C-terminal domain (203-428 aa). By using AlphaFold-Multimer[17], we predicted heteromeric interfaces among M$^4$-FgTad2, FgTad3, and Ame1, showing that the ternary complex contained two semi-independent domains. The first domain comprised mainly FgTad2 and the FgTad3 C-terminal domain, while the second domain comprised mainly Ame1 and the FgTad3 N-terminal domain (Fig. 4j). Contacting Ame1 to the N-terminal domain of FgTad3 could potentially alter the substrate recognition and specificity of the deaminase, given that the FgTad3 N-terminal domain is recognized for its tRNA binding capabilities[5-7].

## Evolution of Ame1 and its key residues for mRNA editing

Ame1 is a protein of unknown function that contains a DUF726 domain belonging to the alpha/beta hydrolase superfamily (cl21494) (Fig. 5a). It also possesses the catalytic triad (S-D-H) conserved in other members of this superfamily. We constructed mutants in PH-1 to investigate whether the function of Ame1 on mRNA editing depends on the catalytic triad. The *AME1$^{DS89A}$* mutant formed small perithecia resembling those of the *ame1/ssc20* deletion mutant, However, both the *AME1$^{SS31A}$* and *AME1$^{SS31A,H629A}$* mutants showed normal sexual development (Fig. 5b). Y2H assays revealed that Ame1$^{D589A}$ did not interact with FgTad3 (Fig. 5c). Consequently, while the residue D$^{589}$ plays a crucial role in the interaction with FgTad3, the catalytic triad is not essential for Ame1's function in mRNA editing.

Ame1 orthologs are widely distributed in ascomycetes but have been lost in Saccharomycotina (Fig. 5a). Orthologs have also been

detected in Agaricomycetes and Chytridiomycetes. Multiple independent gene duplication and horizontal gene transfer events have occurred in different lineages in filamentous ascomycetes, with an ancestral duplication event giving rise to two copies in the last common ancestor of Sordariomycetes and Leotiomycetes (Fig. 5d). The branch length of Ame1 in Sordariomycetes was notably longer than that in Leotiomycetes (Fig. 5a), indicating accelerated evolution of Ame1 in Sordariomycetes. As A-to-I mRNA editing is a common feature in Sordariomycetes but has not been found in Leotiomycetes yet, we hypothesized that the function of Ame1 in mRNA editing has evolved in Sordariomycetes. To test this hypothesis, we replaced the *AME1* ORF in situ in PH-1 with the ORF of the *AME1* ortholog *SsAME1* from *Sclerotinia sclerotiorum*, a species in Leotiomycetes. The *ame1/SsAME1* transformants exhibited similar defects to those of the *ame1* deletion mutant (Fig. 5b), indicating that *AME1* cannot be replaced by *SsAME1*. The Y2H assay showed that SsAme1 did not interact with FgTad3 and SsTad3, but Ame1 did interact with SsTad3 (Fig. 5c). This indicates that the ability of Ame1 to interact with Tad3 evolved in Sordariomycetes.

We identified three amino acid sites that may potentially contribute to the functional innovation of Ame1 in Sordariomycetes (Fig. 5e). We created mutants for these sites in situ in PH-1 by replacing the conserved residues in Sordariomycetes with those in Leotiomycetes. The *AME1$^{V408T}$* mutant displayed normal phenotypes, while the *AME1$^{MV500-501KN}$* mutant exhibited similar defects to the *ame1* deletion mutant (Fig. 5b), indicating that MV$^{500-501}$ is essential for Ame1's function in mRNA editing. Based on the protein structural model, the MV$^{500-501}$ residues are located at the interaction interface between Ame1 and the N-terminal domain of FgTad3 and directly contact the interface residues of FgTad3 (Fig. 5f). Y2H assays confirmed that MV$^{500-501}$ is crucial for the interaction of Ame1 with FgTad3 (Fig. 5c). Because the ancestors of Ame1 had KN residues at this position, the replacement of KN with MV in Sordariomycetes was a critical step in the evolution of Ame1's ability to interact with Tad3 and enable mRNA editing.

## Amino acid residues in FgTad3 exclusively crucial for mRNA editing

To identify the amino acid residues of FgTad3 that are uniquely important for mRNA editing, we engineered FgTad3 variants utilizing the repeat-induced point mutation (RIP) mechanism, which induces C-to-T mutations randomly in repetitive sequences prior to meiosis[18]. The coding region of *FgTAD3* without the start codon was inserted in a reverse orientation before the promoter region of *FgTAD3* in situ, aiming to generate tandemly repeated *FgTAD3* without affecting the transcription of the native gene (Fig. 6a). The resulting TR-*FgTAD3* transformants exhibited normal growth. However, during sexual reproduction, a few abnormal ascospores were observed (Fig. 6b), indicating that RIP mutations had occurred in *FgTAD3*. We isolated ascospores with abnormal morphology at 10 dpf and obtained a total of 350 ascospore progeny. Among these, 33 showed normal growth but exhibited varying degrees of defects in sexual development (Supplementary Data 2). As the inserted *FgTAD3* segment is not transcribed, we amplified and sequenced the native *FgTAD3* gene. All ascospore progeny had at least one RIP mutation resulting in amino acid changes in FgTad3. A total of 52 non-synonymous RIP mutation sites were identified, with most of them located in the N-terminal part of FgTad3 (Supplementary Data 2).

Among the ascospore progeny, seven showed normal growth but severe defects in sexual development (Fig. 6b), each containing a single non-synonymous RIP mutation in *FgTAD3*. Therefore, the observed defects can be attributed to the specific mutational effects. Out of these progeny, progeny 3-2-29 (H343Y) displayed defects only in ascospore formation. The other six progeny produced small perithecia, but no asci were observed in the perithecia formed by progeny 3-19-41 and 3-19-86 (both carrying the H114Y mutation), 3-2-37 (H130Y),

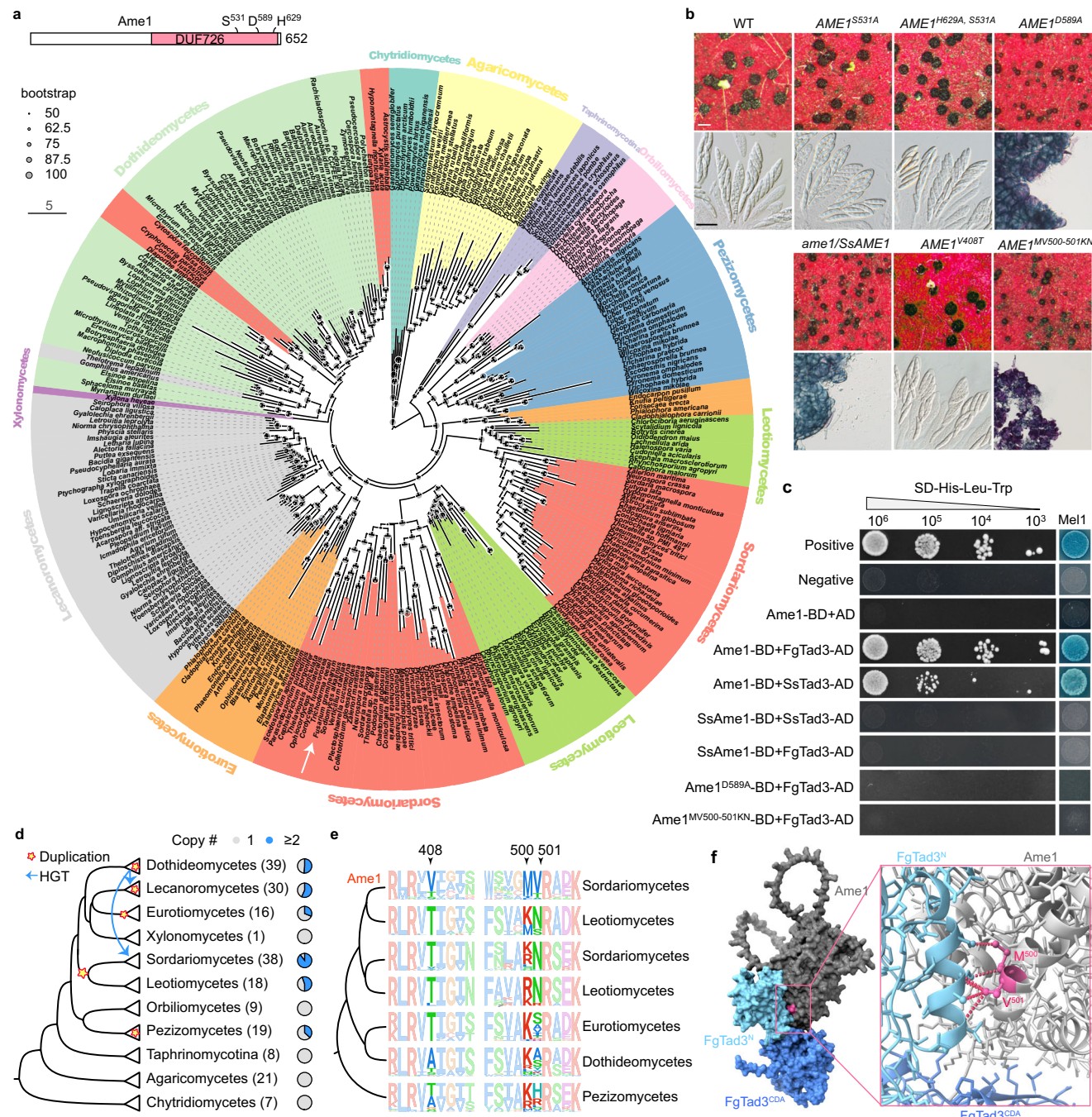

**Fig. 5 | Evolutionary origin of Ame1 and its key residues for mRNA editing in Sordariomycetes. a** The protein domain of Ame1 and a Maximum Likelihood phylogenetic tree of Ame1 orthologs from different fungal lineages. The putative catalytic triad (S·D·H) active sites are indicated. The white arrow indicates the position of *AME1* in the tree. **b** Morphology of perithecia and asci/ascospores of different *AME1* mutants. White bar = 0.2 mm; black bar = 20 μm. **c** Y2H assays for the interaction among Ame1, Ame1 mutants, FgTad3, and their orthologs SsAme1 and SsTad3 from *Sclerotinia sclerotiorum*. **d** The dendrogram of different fungal taxa showing evolutionary events related to Ame1 orthologs.

The number of species in each taxon examined is indicated in the bracket. Each pie chart shows the proportion of fungal species with 1 or ≥ 2 copies of *AME1* orthologs derived from gene duplication and/or horizontal gene transfer events. The putative origin of gene duplication and horizontal gene transfer events are indicated in the dendrogram. **e** WebLogo illustrating the conservation of amino acid residues within different fungal taxa at three putative sites responsible for the mRNA editing ability of Ame1. **f** The protein model showing the direct contact between the MV$^{500-501}$ residues of Ame1 and the interface residues of FgTad3.

and 3-19-3 (M359I). Only a few elongated asci without ascospores were observed in the perithecia formed by progeny 3-19-92 and 3-2-6 (both carrying the P178L mutation). Notably, among the progeny analyzed, the M120I mutation was the most frequent, observed in 9 progeny, which also carried additional mutations. We generated the M120I mutation in situ in PH-1. The resulting *FgTAD3$^{M120I}$* mutant exhibited normal growth but formed small perithecia without ascogenous

hyphae (Fig. 6b). Only 40 A-to-I mRNA editing sites were identified in 7-dpf perithecia of *FgTAD3$^{M120I}$* mutants (Supplementary Table 1), confirming the critical role of M$^{120}$ in mRNA editing. Interestingly, 6 of the progeny with the M120I mutation exhibited small perithecia without ascogenous hyphae, similar to what was observed in *FgTAD3$^{M120I}$* mutants (Fig. 6b). However, a few ascogenous hyphae could be visible inside the slightly small perithecia formed by progeny

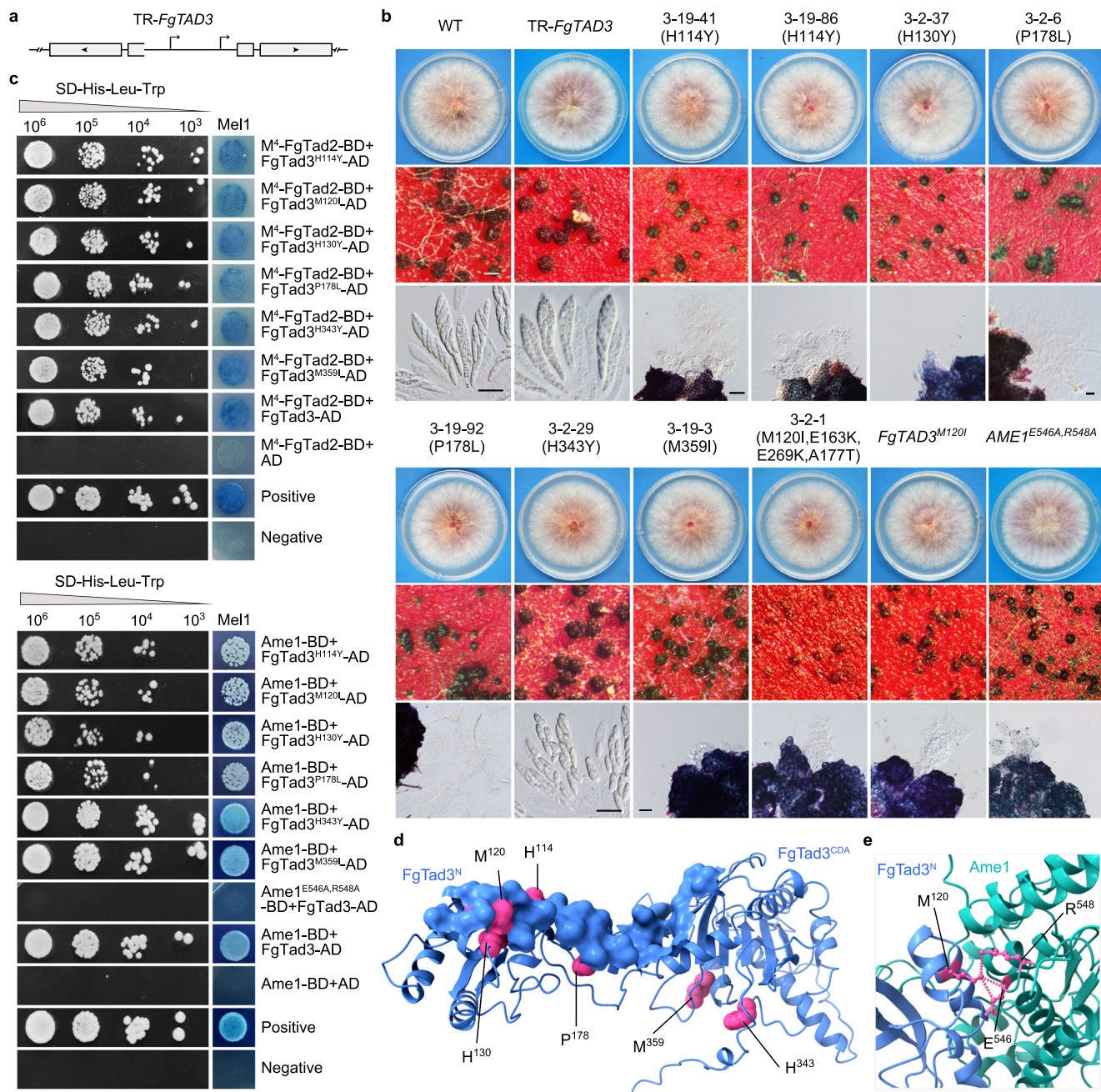

**Fig. 6 | Amino acid residues in FgTad3 exclusively crucial for sexual reproduction and interaction with Ame1. a** Schematic representation of tandemly repeated *FgTAD3* construct in TR-*FgTAD3* strain used for RIP mutations of *FgTAD3*. Arrows indicate transcriptional initiation sites and boxes indicate CDS regions. **b** Morphology of colonies, perithecia, and asci/ascospores of the TR-*FgTAD3* strain, its ascospore progeny with RIP mutations, and the marked mutants of *FgTAD3* and *AME1*. White bar = 0.2 mm; black bar = 20 μm. **c** Y2H assays for the interaction of the marked mutant alleles. **d** Location of mutated residues in proximity to contacting surface of FgTad3 with Ame1. **e** The protein model showing the direct contact between the Ame1 $E^{546}R^{548}$ residues and the FgTad3 $M^{120}$ residue.

3-2-50, 3-19-94, and 3-19-50, which all contain an additional V82I mutation (Supplementary Data 2). These results suggest that the V82I mutation may have a compensatory effect on the M120I mutation.

Y2H assays showed that the mutant alleles FgTad3$^{H114Y}$, FgTad3$^{M120I}$, FgTad3$^{H130Y}$, and FgTad3$^{P178L}$ had a normal interaction with M$^4$-FgTad2 but a decreased interaction with Ame1(Fig. 6c), suggesting that these sites in FgTad3 are important for the interaction with Ame1. Consistently, residues H$^{114}$, M$^{120}$, H$^{130}$, and P$^{178}$ are located within or close to the interacting surface of FgTad3 and Ame1 (Fig. 6c). On the other hand, the interaction with both Ame1 and FgTad2 did not show any obvious changes for FgTad3$^{H343Y}$ and

FgTad3$^{M359I}$ (Fig. 6c). The H343Y and M359I mutations are located within the FgTad3 CDA domain and could potentially impact the structure of the catalytic domain essential for accommodating the stem-loop structures of mRNA.

We identified two residues, E$^{546}$ and R$^{548}$, in Ame1 that were directly in contact with M$^{120}$ of FgTad3 (Fig. 6e). To investigate the role of these residues, we introduced E546A and R548A mutations in situ in PH-1. The *AME1$^{E546A,R548A}$* mutant exhibited normal growth but formed small perithecia without ascogenous hyphae (Fig. 6b). There was no interaction observed between Ame1$^{E546A,R548A}$ and FgTad3, highlighting the critical role of E$^{546}$R$^{548}$ in the interaction between Ame1 and FgTad3.

## The complex edits mRNA in yeasts, bacteria, and human cells

To investigate the editing activities of the FgTad2-FgTad3-Ame1 complex in heterologous systems, we co-expressed M⁴-*FgTAD2*, *FgTAD3*, and *AME1* in the *S. cerevisiae* strain, INVSc1 under the control of the *GAL1* promoter. As controls, we also co-expressed M⁴-*FgTAD2* and *FgTAD3* or expressed only *AME1*, as well as co-expressed *FgTAD3*, *AME1*, and a mutant allele M⁴-*FgTAD2* with the active site E[121] mutated into A (M⁴-*FgTAD2$^{E121A}$*). The A-to-I mRNA editing sites that naturally occur in *FgTAD3* and *AME1* were used as targets to assess the editing activity. By analyzing the RT-PCR products amplified from the total RNA isolated from the yeast transformant co-expressing M⁴-*FgTAD2*, *FgTAD3*, and *AME1*, we observed two peaks (A and G) at the target editing sites in Sanger sequencing traces (Fig. 7a). In contrast, only one A peak was detected at the target sites in the yeast transformant co-expressing M⁴-*FgTAD2$^{E121A}$*, *FgTAD3*, and *AME1*, co-expressing M⁴-*FgTAD2* and *FgTAD3* or expressing only *AME1*. These results indicate that the FgTad2-FgTad3-Ame1 complex can perform A-to-I mRNA editing at the target sites in yeast, and the E[121] in *FgTAD2* is crucial for editing. We also tested the editing activity of this complex in human HEK 293T cell lines. A-to-I mRNA editing was detected at the target sites of *FgTAD3* and *AME1* in the HEK 293T cells, with high editing activities. In contrast, editing was not detected at the target sites in the cells co-expressing M⁴-*FgTAD2* and *FgTAD3* or expressing only *AME1*. These results suggest that the FgTad2-FgTad3-Ame1 complex has universal activity in eukaryotic cells. Furthermore, we tested the editing activity of this complex in the *E. coli* BL21 strain using codon-optimized sequences. A-to-I mRNA editing was detected at a site in the *FgTAD3* transcript in the BL21 transformant co-expressing all three genes but not in the transformant co-expressing only M⁴-*FgTAD2* and *FgTAD3*. This suggests that the FgTad2-FgTad3-Ame1 complex also has mRNA editing activity in prokaryotic cells.

To evaluate the editing of endogenous transcripts in the INVSc1 and BL21 strains, we conducted DNA-seq and strand-specific RNA-seq analyses. We identified a total of 1114 and 1241 A-to-I mRNA editing sites in the endogenous mRNAs in the INVSc1 and BL21 transformants co-expressing all three genes (Fig. 7b; Supplementary Fig. 10), respectively. Notably, the editing levels of the detected editing sites in the BL21 transformants were significantly higher than those detected in the INVSc1 transformants (Fig. 7c). We observed similar nucleotide preference surrounding the editing sites and enrichment of editing sites in hairpin-loop structures in both INVSc1 and BL21 transformants (Fig. 7d, e), as previously reported in *F. graminearum* [12]. Therefore, the FgTad2-FgTad3-Ame1 complex exhibits robust A-to-I mRNA editing activities in heterologous systems.

## Discussion

A-to-I RNA editing systems have garnered attention for their mRNA-level genetic information recoding ability. This allows for correcting pathogenic mutations and genetic information reprogramming, which is particularly relevant as C:G to T:A genetic mutations account for approximately half of all known pathogenic point mutations in humans[19,20]. Site-specific RNA editing systems utilizing ADARs have been developed to target disease-causing mutations[21,22]. TadA has been associated with a small number of naturally occurred A-to-I mRNA editing sites in bacteria[23,24]. Overexpression of TadA-derived adenine base editors can induce widespread A-to-I editing in cellular mRNAs[25]. However, the eukaryotic tRNA-specific heterodimeric deaminase Tad2-Tad3 typically requires the complete tertiary structure of cognate tRNAs for deamination and is not known for mRNA editing capability[26,27]. Our recent results indicate that FgTad2 catalyzes A-to-I mRNA editing in *F. graminearum*[28]. This study unveils the fungal A-to-I mRNA editing machinery and demonstrates that the Ame1 protein

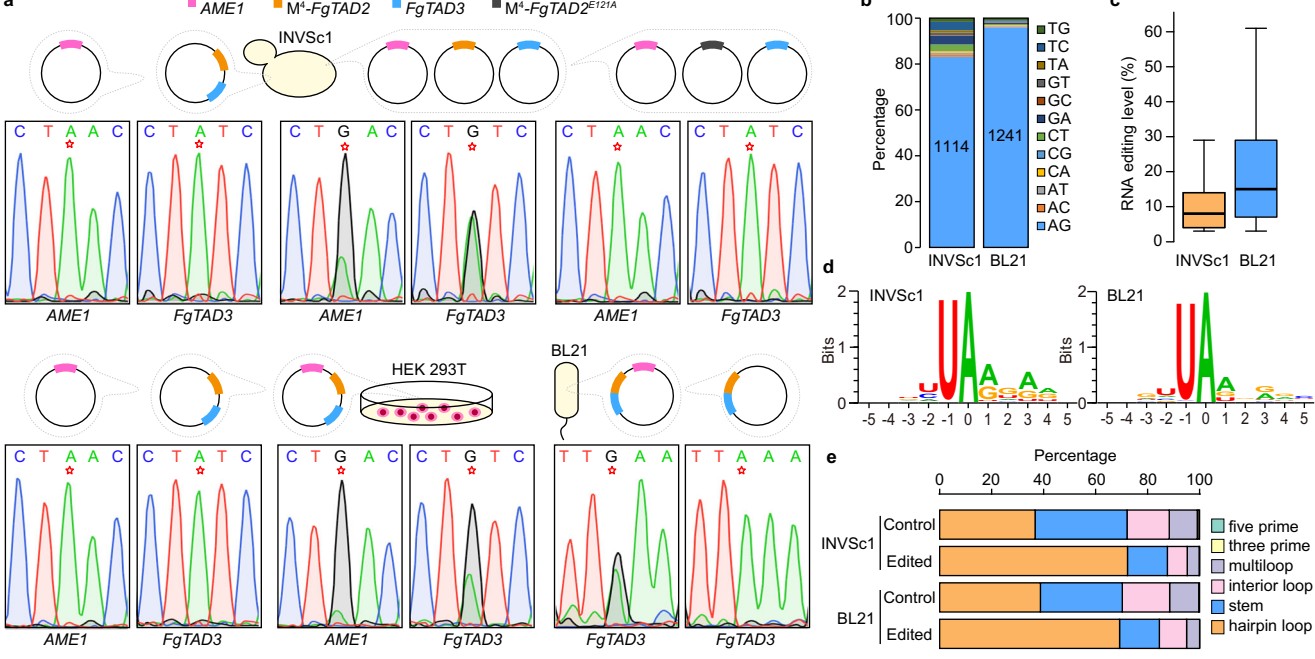

**Fig. 7 | A-to-I mRNA editing activities of the FgTad2-Fg-Tad3-Ame1 complex in heterologous systems. a** Schematic representation of the expression system used for transformation of the yeast INVSc1 strain, the *E. coli* BL21 strain, and the human HEK 293T cell line. The A-to-I mRNA editing sites in *FgTAD3* and *AME1* were used as targets to assess the editing activity using Sanger sequencing of the RT-PCR products. The target editing sites are marked by a red star. When editing occurs, two peaks (A and G) at the editing sites are observed in the sequencing traces. **b**, **c** The number and editing level (%) of detected A-to-I RNA editing (AG) sites in the yeast INVSc1 strain

(*n* = 1114) and *E. coli* BL21 strain (*n* = 1241) expressing the FgTad2-FgTad3-Ame1 complex. Boxplots indicate median (middle line), 25th, 75th percentile (box), and 1.58x interquartile range (whisker). **d** WebLogo showing base preferences in the flanking sequences of the editing sites. **e** Stacked columns showing fractions of different types of RNA secondary structure elements predicted based on 30-nt upstream and 30-nt downstream sequences of the edited A sites. For control, an equal number of A sites with similar nucleotide preference at the −2 to +4 positions were randomly selected from transcript sequences. Source data are provided as a Source Data file.

enables FgTad2-FgTad3 to conduct transcriptome-wide A-to-I mRNA editing by interacting with the N-terminal domain of FgTad3 (Fig. 8). The ability of the fungal editing machinery to efficiently edit mRNA in yeasts, bacteria, and human cells, makes it a promising tool in therapy and agriculture. The fungal editing system and ADARs, which have different substrate preferences, can complement each other in RNA editing applications. Additionally, the preference of the fungal editing machinery for the UAG triplet[8] makes it especially useful for correcting nonsense mutations, which are responsible for a significant proportion of human inherited diseases[29].

While other regulatory factors may contribute to the editing efficiency, our findings suggest that the FgTad2-FgTad3-Ame1 ternary complex is sufficient for A-to-I editing of mRNA. The Ame1-FgTad3 interaction might induce a conformational change in the FgTad2-FgTad3 complex, potentially diminishing the specificity for recognizing substrate RNA structures. This interaction broadens the substrate range of FgTad2-FgTad3 from tRNA to encompass mRNA. Previous reports have shown that altering specific residues through directed evolution can modify the substrate recognition and specificity of a deaminase[30-32]. Our work provides a perspective on how to control the substrate recognition and specificity of editing enzymes by utilizing their interaction with other partners. Furthermore, we have identified specific amino acid mutations in FgTad3 that exclusively impact mRNA editing. This helps to elucidate the differences in the mechanisms of tRNA- and mRNA-substrate recognition by this complex.

Our study has demonstrated that Ame1 is specific to the sexual stage and plays a direct role in the sexual stage-specific activity of A-to-I mRNA editing. The resemblance in phenotypes between the *ame1* deletion mutant and the uneditable mutant of *PSC58*[14] implies

that Ame1's principal function likely lies in mRNA editing, although the possibility of other functions cannot be excluded. Despite Ame1's classification in the alpha/beta hydrolase superfamily, its role in mRNA editing appears independent of the putative catalytic activity of alpha/beta hydrolases. We found that while A-to-I mRNA editing is essential for sexual development, its activation during vegetative growth can have harmful effects. This implies that edited versions of certain genes are advantageous in sexual stages but disadvantageous in vegetative stages. Hence, enhancing sexual reproduction through genomic mutations may incur reproductive costs. The formation of the FgTad2-FgTad3-Ame1 complex specific to the sexual stage may confer an adaptive advantage by circumventing trade-offs between survival and reproduction. *F. graminearum* is a major cause of Fusarium head blight (FHB), one of the most devastating diseases on wheat and barley worldwide. Ascospores discharged from perithecia are the primary inoculum of FHB[33]. Targeting the fungi-specific RNA editing machinery will be a promising approach to controlling fungal plant pathogens that rely on ascospores for infection[34,35].

In addition to the sexual stage-specific interaction with Ame1, we have discovered that FgTad2-FgTad3 is regulated by alternative promoters, alternative translation initiation, and post-translational modifications (Fig. 8). This indicates that the expression and activity of FgTad2-FgTad3 are tightly controlled within the cell. Notably, the M⁴-FgTad2 protein encoded by the sexual stage-specific S-transcript exhibits higher editing activity due to a stronger interaction with FgTad3 compared to the M¹-FgTad2 protein encoded by the L-transcript. Surprisingly, the M⁴-FgTad2 isoform is also produced by the L-transcript during growth through alternative translation initiation. According to the scanning mechanism of eukaryotic

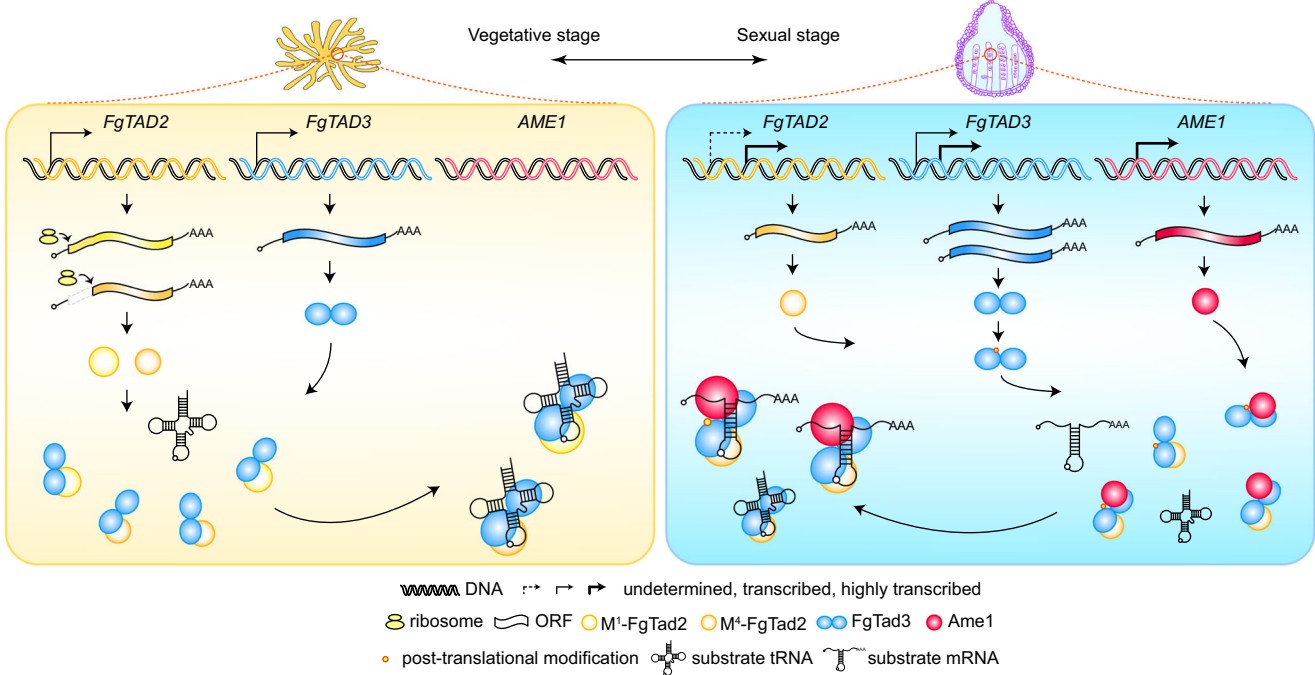

**Fig. 8 | Proposed models for components, expression, and regulation of A-to-I mRNA editing machinery in *F. graminearum*.** *FgTAD2* encodes proteins that contain a CDA domain, while *FgTAD3* encodes proteins that contain both an N-terminal RNA-binding domain and a C-terminal CDA domain. FgTad2 interacts with the CDA domain of FgTad3 to form the catalytic domain responsible for A-to-I editing of both tRNAs and mRNAs. During vegetative stages, *FgTAD2* expresses two protein isoforms: M¹-FgTad2 and M⁴-FgTad2, via alternative translational initiation. Compared to M¹-FgTad2, M⁴-FgTad2 has a stronger interaction with FgTad3. In sexual stages, both *FgTAD2* and *FgTAD3* generate a short transcript isoform via

alternative transcriptional initiation in fertile tissues. The long transcript isoform of *FgTAD3* is also expressed in fertile tissues, while that of *FgTAD2* remains to be determined. The short transcript isoform of *FgTAD2* encodes proteins identical to M⁴-FgTad2, while both transcript isoforms of *FgTAD3* encode the same proteins. A gene named *AME1* (activator of mRNA editing) is induced to expression in sexual stages, which encodes proteins with a DUF726 domain. Ame1 interacts with the N-terminal domain of FgTad3 to mediate the recognition of mRNA substrates. The FgTad2-FgTad3-Ame1 complex performs A-to-I mRNA editing. FgTad3 also undergoes post-translational modification that is important for mRNA editing.

translation initiation[36], the M[1] codon of the *FgTAD2* L-transcript may not be in an optimal context; inefficient recognition of this codon can lead to the initiation of translation at the optimized M[4] codon. Additionally, the M[1]-FgTad2 isoform has an unusually long N-terminal part compared to orthologs in yeast, plants, and animals, suggesting that M[4]-FgTad2 represents the ancestral state. It is known that long 5'UTRs tend to exhibit lower translation efficiency[37]. Since randomly induced mRNA editing activity is not inherently specific to sexual reproduction initially, we hypothesize that the L-transcript of *FgTAD2* and *FgTAD3* may have evolved as an acquired characteristic to alleviate the adverse effects of mRNA editing during vegetative stages.

Our study has revealed that the role of Ame1 in mRNA editing has evolved in Sordariomycetes, despite the presence of Ame1 orthologs in various fungal lineages. We have identified the key residues in Ame1 responsible for the innovation of mRNA editing in Sordariomycetes. Interestingly, sexual stage-specific A-to-I mRNA editing has also been identified in *Pyronema confluens* (*omphalodes*)[38], an early-diverging filamentous ascomycete belonging to Pezizomycetes. Our results suggest that A-to-I mRNA editing in the two fungal lineages has independent origins. Notably, *P. confluens* and its closely related lineages in Pezizomycetes also contain an additional duplicated copy of Ame1 orthologs, which raises the possibility that A-to-I mRNA editing in Pezizomycetes has also emerged due to the duplication of Ame1 orthologs. This work provides a strong motivation for future investigations of A-to-I mRNA editing in fungal lineages with gene duplication and horizontal gene transfer events in Ame1 orthologs.

## Methods

### Strains and cultural conditions
The *F. graminearum* wild-type strain PH-1[39] and its derived mutants and transformants were routinely cultured on potato dextrose agar (PDA) plates (20% potato, 2% dextrose, and 1.5% agar) at 25 °C. Transformants were generated using PEG-mediated transformation of protoplasts[40]. For transformant selection, hygromycin B (H005, MDbio, China) and geneticin (Sigma-Aldrich, St. Louis, MO) were added to the final concentration of 300 μg/mL and 200 μg/mL, respectively, in Top agar medium (0.3% yeast extract, 0.3% casamino acids, 20% sucrose, and 1.5% agar). The growth rate and colony morphology were assayed on PDA plates for three days. Vegetative hyphae used for DNA and RNA isolation were harvested from 24 h liquid YEPD cultures (0.3% yeast extract, 1% peptone, and 2% dextrose). Conidia were harvested from 5-day-old liquid carboxymethyl cellulose (CMC) cultures (1.5% carboxymethylcellulose, 0.1% $NH_4NO_3$, 0.1% $KH_2PO_4$, 0.05% $MgSO_4 \cdot 7H_2O$, and 0.1% yeast extract). For self-fertilization of sexual reproduction, aerial hyphae on 6-day-old carrot agar plates were pressed down with 500 μL of sterile 0.1% Tween-20 and subsequently cultured under black light at 25 °C. For outcrosses, the female strains grown on carrot agar plates were fertilized with 1 mL of conidial suspension ($1 \times 10^6$ conidia/mL) derived from the male strains. Perithecium formation and cirrhi production were examined using an Olympus SZX16 stereoscope. Ascogenous hyphae, asci, and ascospores were observed with an Olympus BX-51 microscope in squash mounts of perithecia 7 days post-fertilization (dpf). For scanning electron microscopy, perithecia produced on wheat straw were coated with gold-palladium and examined with a JEOL 6360 scanning electron microscope (Jeol).

### Gene deletion and complementation
Gene deletion was performed using the split-marker approach[41]. Fragments of approximately 1.0 kb upstream and downstream of the target genes were amplified and connected to the N- and C-terminal regions of either a hygromycin-resistance cassette or a recyclable marker module[12] that confers resistance to hygromycin and sensitivity to the nucleoside analog 5-fluoro-2'-deoxyuridine (Floxuridine)

through overlapping PCR. After transforming PH-1 protoplasts, hygromycin-resistant transformants were screened and confirmed by PCR assays. At least two independent deletion mutants were obtained for each gene. For gene complementation, the full-length genes, including their native promoter, were amplified and cloned into a pFL2 vector using the yeast gap-repair approach[42]. The complementation construct was confirmed by DNA sequencing and then transformed into the protoplasts of the corresponding deletion mutants. Transformants containing the complementation constructs were screened with geneticin and confirmed by PCR assays. Supplementary Data 3 lists all the strains used in this study, while Supplementary Data 4 lists the primers used.

### Overexpression of *FgTAD2* and *FgTAD3* in *F. graminearum*
To overexpress the *FgTAD2* and *FgTAD3* genes in a target locus, the *FG1G36140* deletion mutant was used, which was generated by the recyclable marker module and had indistinguishable phenotypes from the wild type. The coding regions of *FgTAD2* or *FgTAD3* were PCR amplified and connected to the constitutive RP27 promoter from the pFL2 vector through overlapping PCR. Fragments of approximately 1.0 kb upstream and downstream of the *FG1G36140* gene were PCR amplified and connected to the $P_{RP27}$-*FgTAD2* or $P_{RP27}$-*FgTAD3* fragments through overlapping PCR. The PCR products were confirmed by sequencing analysis and then transformed into protoplasts of the *FG1G36140* deletion mutant. Transformants resistant to 25 μg/mL Floxuridine (HY-B0097, MCE, USA) were screened. To overexpress *FgTAD3* or *FgTAD2* initiated by M[1] or M[4] codons in the *FG1G36140* locus of *AME1*-oe transformants, two DNA fragments were generated using overlapping PCR. One DNA fragment contained the upstream homologous fragment of *FG1G36140*, along with either the $P_{RP27}$-*FgTAD3*, $P_{RP27}$-M[1]-*FgTAD2*, or $P_{RP27}$-M[4]-*FgTAD2* fragment, and the N-terminal region of the geneticin-resistance cassette. The other DNA fragment contained the C-terminal region of the geneticin-resistance cassette and the downstream homologous fragment of *FG1G36140*. These two DNA fragments were then co-transformed into the protoplasts of the *AME1*-oe transformants. Transformants resistant to geneticin were screened and confirmed by PCR assays.

### Site-directed mutagenesis and allelic exchanges
Allelic fragments with desired mutations, deletions, or insertions were generated using overlapping PCR. For allelic exchanges at the native gene locus, the allelic fragments were connected to the N-terminal region of the hygromycin-resistance cassette through overlapping PCR. Fragments of approximately 1.0 kb downstream of the target regions were amplified and connected to the C-terminal region of the hygromycin-resistance cassette, also through overlapping PCR. Both fragments were co-transformed into PH-1 protoplasts. Transformants resistant to hygromycin were screened by PCR, and desired mutations were confirmed by DNA sequencing. Quantitative PCR was used to confirm that the transformants did not have any unintended integration of allelic fragments. For each mutation, at least two independent mutants with similar phenotypes were identified.

### Identification of repeat-induced point mutations in *FgTAD3*
To generate the tandemly repeated *FgTAD3* in situ, the coding region of *FgTAD3* without the start codon was amplified and inserted in a reverse orientation before the promoter region (1832 bp upstream of the start codon) of *FgTAD3*. Two DNA fragments were generated using overlapping PCR. One fragment contained the upstream homologous fragment, the *FgTAD3* coding region, and the N-terminal region of the hygromycin-resistance cassette. The other fragment contained the C-terminal region of the hygromycin-resistance cassette and the downstream homologous fragment. These two DNA fragments were co-transformed into PH-1 protoplasts. Hygromycin-resistant

transformants were screened and confirmed using PCR assays. Quantitative PCR was performed to ensure that unintended integration of the introduced fragments did not occur in the resulting TR-*FgTAD3* transformants.

For single ascospore isolation, ascospore cirrhi were collected from 10 dpf perithecia formed by the TR-*FgTAD3* transformants on carrot agar plates under a dissecting microscope and resuspended in 1 mL of sterile distilled water in a 1.5 mL centrifuge tube. The ascospore suspensions were then spread on 2% water agar, and ascospores with morphological defects were isolated using single spore isolation[43], and transferred onto PDA plates. The isolated single ascospores were then assayed for defects in growth and sexual reproduction. To identify mutations in the ascospore progeny, the endogenous *FgTAD3* gene was amplified from the DNA isolated from ascospore progeny and sequenced using Sanger sequencing.

## 5′RACE and RT-PCR

Total RNA was extracted from hyphae or perithecia using the TRIzol reagent (Invitrogen, USA). The Rapid Amplification of 5′-Ends cDNA (5′RACE) assays were performed using the FirstChoice® RLM-RACE Kit (Invitrogen, USA) following the manufacturer's instructions. In brief, mRNAs were converted into first-strand cDNA using reverse transcriptase Super Script™ II and a gene-specific primer 1 (GSP1). Following cDNA synthesis, the first-strand product was purified to remove unincorporated dNTPs and GSP1. Terminal deoxynucleotidyl transferase (TdT) was used to add homopolymeric tails to the 3′ ends of the cDNA. Tailed cDNA was then amplified by PCR using a gene-specific primer 2 (GSP2) downstream of GSP1 and an adapter primer that permitted amplification from the homopolymeric tail. The PCR products were detected by electrophoresis and connected to a T-vector for Sanger sequencing analysis.

For RT-PCR, cDNA synthesis was performed using RevertAid Master Mix (Thermo Scientific) following the manufacturer's instructions. Reverse transcription (RT)-PCR products were gel-purified and subjected to direct sequencing. Sanger sequencing traces were visualized using SnapGene Viewer 4.3 (https://www.snapgene.com/snapgene-viewer/). Relative expression levels were assayed by quantitative real-time RT-PCR using the $2^{-\Delta\Delta CT}$ method[44], with the *GAPDH* gene serving as an internal control.

## Yeast two-hybrid (Y2H) assays

To detect protein interactions using yeast-two-hybrid (Y2H) assays, the open reading frames (ORFs) of genes and their mutant alleles were amplified and cloned into the pGADT7 prey or pGBKT7 bait vector (TaKaRa Bio, Tokyo, Japan). The resulting bait and prey constructs were transformed in pairs into yeast strain AH109 and assayed for growth on SD-His-Leu-Trp plates. Filter papers containing X-gal were used to assess the activity of LacZ β-galactosidase. SD-His-Leu-Trp or SD-His-Leu-Trp-Ade plates containing X-α-gal were used to evaluate the activity of Mel1 α-galactosidase.

## Western blotting

Total proteins were isolated and separated on 10% SDS-PAGE gels and transferred to nitrocellulose membranes. The nitrocellulose membranes were incubated in 5% skim milk for 1 hour to avoid non-specific binding of antibodies. The proteins were then incubated with the anti-GFP (Abcam, ab290), anti-FLAG (Sigma-Aldrich, F31651), or anti-Tub2[45] antibodies overnight at 4 °C. After washing three times with 1×TBST (100 mM Tris-HCl, pH 7.5; 136 mM NaCl; 0.1% Tween-20), the proteins were incubated with corresponding secondary antibodies at room temperature for 1 h. The non-specifically bound secondary antibodies were washed away with 1×TBST, and the proteins were incubated in a chemiluminescent substrate for 5–6 min before being detected by a chemiluminescence system.

## Co-immunoprecipitation (co-IP)

For co-IP to assay the interaction between FgTad2 and FgTad3, the $P_{RP27}$-*FgTAD2*–3×FLAG and $P_{RP27}$-*FgTAD3*-GFP fusion constructs were generated using the yeast gap-repair approach[42] and co-transformed into PH-1. Total proteins were isolated from 24-h hyphae of the resulting transformants. The supernatant containing the target protein was incubated with anti-GFP affinity beads (SA070005, Smart-Lifesciences, China), and proteins were eluted. Western blots were performed on both total proteins and proteins eluted from the anti-GFP affinity beads using anti-FLAG (Sigma-Aldrich, F31651), anti-GFP (Abcam, ab290), or anti-Tub2[45] antibodies for detection.

To investigate the interaction between Ame1 and FgTad3, fusion constructs of *AME1*−6×His and *FgTAD3*-Stag were created by inserting their ORFs into the MCS1 and MCS2 regions of the pRSFDuet-1 vector, respectively, which was then transformed into *E. coli* BL21. The resulting transformants were confirmed via PCR and western blot analysis. For co-IP assays, a 500 μL overnight bacterial culture was added to 50 mL of LB media and grown at 37 °C until reaching an OD$^{600}$ of 0.5-0.8. Recombinant protein expression was induced with isopropyl β-d-1-thiogalactopyranoside (IPTG) at a final concentration of 0.1 mM overnight at 20 °C. Cells were lysed by sonication and the resulting clear lysate was harvested by centrifugation at 5000 rpm for 20 min, then transferred to a 4 mL tube. The supernatant containing the target protein was incubated with Ni-NTA agarose beads (Smart-Lifesciences, China) at 4 °C overnight with gentle rocking for affinity purification. Western blots were performed on both total proteins and proteins eluted from the Ni-NTA beads using anti-His (CW0286M, CMBIO, China), anti-Stag (Cell Signaling Technology, 12774 S), or anti-GAPDH (Sangon Biotech, D1100160200) antibodies.

## Assaying mRNA editing activities in heterologous systems

To express *AME1*, M⁴-*FgTAD2*, and *FgTAD3* in yeast for assaying mRNA editing activities of the FgTad2-FgTad3-Ame1 complex, we amplified their ORFs and cloned them into pYES2, pYES2-Leu, and pYES2-Trp vectors, respectively, under the control of the *GAL1* promoter. The pYES2-Leu and pYES2-Trp vectors were derived from the pYES2 vector by digesting with *Nco*I and *Cla*I and replacing the Ura3 gene with Leu2 amplified from pGADT7 plasmid or Trp1 amplified from pGBKT7 plasmid. We co-transformed the resulting pYES2-*AME1*, pYES2-Leu-M⁴-*FgTAD2*, and pYES2-Trp-*FgTAD3* vectors into yeast INVSc1 strain by heat shock. Transformants were confirmed by PCR and cultured in 10 mL 2% glucose SD-Ura-Leu-Trp liquid medium at 30 °C for 24 h. The cells were harvested by centrifugation and resuspended in 10 mL 2% galactose SD-Ura-Leu-Trp liquid medium at 30 °C for 12 h. We isolated total DNA and RNA from the cells and performed PCR and RT-PCR for *FgTAD3* and *AME1*. The purified PCR and RT-PCR products were subjected to direct Sanger sequencing for detecting editing events at the target A-to-I sites.

To assay mRNA editing activities of the FgTad2-FgTad3-Ame1 complex in bacteria, we cloned the codon-optimized ORFs of *AME1*, M⁴-*FgTAD2*, and *FgTAD3* into the pRSFDuet-1 plasmid under the control of both the T7 promoter and *lac* operon. We inserted *AME1* at the MCS1 region, while M⁴-*FgTAD2* and *FgTAD3* were sequentially inserted at the MCS2 region. We transformed the resulting vector into *E. coli* BL21 and confirmed the transformants by PCR. For control, we also transfected BL21 with the vector containing only M⁴-*FgTAD2* and *FgTAD3*. We cultured the transformants in LB media at 37 °C and induced gene expression with isopropyl β-d-1-thiogalactopyranoside (IPTG) at a final concentration of 0.1 mM overnight at 20 °C. We extracted total DNA and RNA and performed PCR and RT-PCR to assay the editing events in transcripts of *FgTAD3* by Sanger sequencing.

To assay mRNA editing activities of the FgTad2-FgTad3-Ame1 complex in human cell lines, we cloned the ORFs of *AME1*, M⁴-*FgTAD2*, and *FgTAD3* into the pCMV-Blank plasmid under the control of the CMV promoter. We used the generated vector to transfect the HEK

293T cell line (Servicebio, China), respectively. For control, we also transfected the cell lines with the vectors containing M[4]-*FgTAD2* and *FgTAD3* or only *AME1*. After culturing at 37 °C for 48 h, we extracted total RNA and performed RT-PCR to assay the editing events in transcripts of *FgTAD3* and *AME1* by Sanger sequencing.

## Identification of post-translational modifications in FgTad3

We amplified the coding region of *FgTAD3* by PCR and cloned it into the vector pDL2 using the yeast gap-repair approach[42]. The ORF of *FgTAD3* was fused with GFP and under the control of the RP27 promoter. The resulting *FgTAD3*-GFP fusion constructs were confirmed by Sanger sequencing analysis and transformed into PH-1. We confirmed hygromycin-resistant transformants expressing the fusion constructs by PCR and western blot analysis.

To identify modifications on FgTad3, we performed tandem mass spectrometry experiments three times. In brief, total proteins were extracted from hyphae harvested from 24 h YEPD cultures and perithecia collected from mating plates at 7 dpf. The proteins were then incubated with anti-GFP affinity beads (SA070005, Smart-Lifesciences, China) at 4 °C for 2.5 h. Following this, the proteins were eluted from the anti-GFP beads and detected by Coomassie blue staining. The FgTad3-GFP band was excised into 1 mm³ squares from the SDS-PAGE gel and subsequently treated with 50 mM $NH_4HCO_3$/30% ACN for bleaching. The transparent gel particles, treated with 10 mM Dithiothreitol (DTT) and 60 mM iodoacetamide (IAM), were digested with trypsin (Promega, USA) in gel at 37 °C for 12 h to cleave the protein at the carboxyl side of lysine and arginine residues. Peptides extracted from gel particles were purified and analyzed by high-sensitivity LC-MS/MS (QExactive HF-X, ThermoFisher, Waltham, USA) to examine the charge and peak diagram of protein fragmentation. The following conditions were used: Mobile phase A consisted of 0.1% formic acid, while mobile phase B consisted of 0.1% formic acid and 80% ACN. The method duration was 140 min, and the flow rate was set at 400 nL/min.

The obtained data were analyzed using Thermo Scientific Proteome Discoverer 2.4 software. The minimum precursor mass was 350 Da, and the maximum was 5000 Da. The minimum peptide length was 6. The desired false discovery rate (FDR) for the strict criteria was set at 0.01, while for the relaxed criteria, the target FDR was 0.05. The validation was based on $q$-value, and the peptides confidence was high. Post-translational modifications such as acetylation, ubiquitination, methylation, and phosphorylation were searched on the protein sequence of FgTad3. We screened for modification sites that appeared more than twice in independent experiments.

## DNA-seq and strand-specific RNA-seq

DNA was extracted using the CTAB method. RNA was extracted and purified using the Eastep Super Total RNA Extraction Kit (Promega, USA). For eukaryotic RNA samples, poly (A)⁺ mRNA was enriched using magnetic beads with Oligo (dT), while for prokaryotic samples, rRNA was removed to enrich mRNA. DNA-seq and strand-specific RNA-seq libraries were constructed and sequenced on an Illumina NovaSeq 6000 system with 2 × 150-bp paired-end read mode at Novogene Bioinformatics Institute (Tianjin, China). Low-quality reads and reads containing adapters were removed by Trimmomatic[46] with default settings. The information regarding the DNA- and RNA-Seq data generated in this study has been listed in Supplementary Data 5.

## RNA immunoprecipitation sequencing (RIP-seq)

We generated a strain expressing *FgTAD2*−3×FLAG at the native locus using the allelic exchange strategy described above. We collected 7-dpf perithecia produced by the *FgTAD2*−3×FLAG strain. RIP-seq assays were conducted with assistance from Wuhan IGENEBOOK Biotechnology (www.igenebook.com). In brief, the perithecia sample was cross-linked

at 400 mj/cm² at 4 °C. Total proteins were then isolated, and 50 µL magnetic beads and 5 µg anti-Flag antibodies were mixed for 30 min at room temperature. The magnetic beads bound with antibodies were incubated with 900 µL total proteins overnight at 4 °C for affinity purification. The FgTad2-3×FLAG proteins were eluted from magnetic beads for 30 min at 55 °C. Input RNA and the RNA extracted from RIP eluate were used to construct sequencing libraries, respectively. The libraries were constructed with the NEBNext Ultra RNA Library Prep Kit following the manufacturer's instruction and sequenced on an Illumina NovaSeq 6000 system with 2 × 150-bp paired-end read mode at Novogene Bioinformatics Institute (Tianjin, China).

## Analysis of RNA-seq and RIP-seq data

We obtained the reference genomes and annotation files of *F. graminearum* PH-1, *S. cerevisiae* R64-1-1, and *E. coli* BL21 from FgBase (http://fgbase.wheatscab.com/)[16], Ensembl Fungi, and NCBI genome database, respectively. The published strand-specific RNA-seq data[8,13] of 24 h vegetative hyphae and 8 dpf perithecia, along with samples from sexual development spanning 1- to 8-dpf, were obtained from the NCBI SRA database. Details regarding these RNA-Seq datasets have been documented in Supplementary Data 5. After adapter trimming, the DNA-seq and RNA-seq reads were mapped to the reference genomes using Bowtie 2 v2.5.1[47] and HISAT2 v2.2.1[48] with the two-step model, respectively. Quality control of alignments was performed with Qualimap 2[49]. The number of reads aligned to each gene was calculated using featureCounts[50] and normalized by RPKM (Reads Per Kilobase per Million mapped reads) or TPM (Transcripts Per Million).

For RIP-seq analysis, the normalized read density of each gene was calculated as $Log_2$ (IP_RPKM/Input_RPKM + 1). Based on the known A-to-I mRNA editing sites identified in our previous study[12], all genes expressed during sexual reproduction were classified into edited and unedited groups using a binary approach. Additionally, we further divided the edited genes into high-edited and low-edited groups based on the editing activities (calculated by summing the editing levels of all editing sites) of each gene. We compared the normalized read density between the binary groups.

For A-to-I editing site identification, duplicated reads in the mapped RNA-seq BAM file were removed using the MarkDuplicates from Picard package v2.18.7 (http://broadinstitute.github.io/picard/). The resulting RNA-seq bam files were split into two bam files containing separated sense-strand and antisense-strand read alignments by BamTools v2.5.2[51]. A-to-I mRNA editing sites were identified by REDItools v1.2[52] using matched DNA-seq and RNA-seq data with an editing level cutoff value of 3%[53]. To exclude false positives, all A-to-G sites detected in samples with less than 100 sites were manually inspected using IGV[54].

The base preference surrounding the editing sites was visualized using WebLogo 3[55]. The secondary structures of 30-nt upstream and 30-nt downstream sequences of the editing or control sites were predicted using RNAFold in Viennarna v2.4.18[56]. We used forgi v2.0.3[57] to perform statistics of secondary structure types. The random sampling of control sites with similar base preferences at −2 to +4 positions as edited sites was performed with the scripts developed in our previous study (https://github.com/wangqinhu/NC.edits)[11].

## Homology searches, phylogenetic analysis, and protein modeling

We identified gene homologs by performing a BLASTp search in the NCBI nr database and identified protein-conserved domains using NCBI CD-Search (https://www.ncbi.nlm.nih.gov/Structure/cdd/wrpsb.cgi). Gene orthologs were identified by orthAgogue[58] with default parameters. Multiple sequence alignments were performed with MUSCLE v5.1[59] using default settings. The phylogenetic tree of Ame1 orthologs was constructed by IQ-TREE v2.2.2.7[60] (-m MFP -T AUTO -B

1000) and visualized by iTOL[61]. Details regarding the Ame1 orthologs used for phylogeny have been documented in Supplementary Data 6. The protein complex structures of FgTad2-FgTad3-Ame1 and FgTad2-FgTad3 were predicted using AlphaFold-Multimer[17] on COSMIC2 (http://cosmic-cryoem.org/). The structures of FgTad2-FgTad3 and tRNA$^{Thr}_{CGU}$ were docked using PRIME v2.0.1[62] with the model (PDB: 8AW3) of ADAT2/3 bound to tRNA from *Trypanosoma brucei*[7] as a template. All protein structures were visualized by ChimeraX v1.4[63]. The eight inosine-modified tRNAs in *F. graminearum* were obtained from the GtRNAdb database (http://gtrnadb.ucsc.edu/).

## Statistics & reproducibility

No statistical method was used to predetermine sample size. The experiments were not randomized. The Investigators were not blinded to allocation during experiments and outcome assessment. No data were excluded from the analyses. We performed statistical significance tests with R. *P*-values were calculated using two-tailed Wilcoxon rank sum tests and adjustments were made for multiple comparisons.

In order to ensure the repeatability of our results, each experiment was independently repeated at least twice, with similar results obtained each time. All the blot images presented in this paper represent consistent results from independent experiments. For phenotype observation, at least two independent mutants/transformants with three biological replicates each were used for each experiment, although only the representative images were shown. For RIP-seq, the results were presented for two independent experiments. In the analysis of RNA editing sites and editing levels by RNA-seq, in most cases, two biological replicates were used. The mean of editing levels calculated from the two replicates was used for statistical comparison of editing levels of shared editing sites in different strains. To facilitate comparison, an equivalent number of deduplicated mapped reads were randomly sampled for the compared strains. These measures ensure that our RNA editing analysis is as robust as possible. Since our aim was to determine whether RNA editing occurred rather than to make comparisons, only one biological replicate was used for RNA-seq of 6-dpf perithecia of the *ssc20/ame1* deletion mutant, 7-dpf perithecia of the *FgTAD3*$^{M120I}$ mutant, and the INVSc1 and BL21 transformants co-expressing *FgTAD2*, *FgTAD3*, and *AME1*.

The published RNA-seq data of *F. graminearum* during sexual development, which were used in this paper, only have one biological replicate for each time point. However, since the primary purpose of RNA-seq data is to illustrate expression and editing profiles rather than to identify significantly differentially expressed genes/transcripts or editing sites, the lack of multiple biological replicates does not affect our conclusions. The consistent expression trends of *FgTAD2* and *FgTAD3* across time points indicate the reliability of the RNA-seq data.

## Reporting summary

Further information on research design is available in the Nature Portfolio Reporting Summary linked to this article.

## Data availability

Data supporting the findings of this work are available within the paper and its Supplementary Information file. Source data are provided with this paper. Omics data generated in this study are accessible under the NCBI BioProject accession: PRJNA1020378. Published omics data used in this study are accessible under the NCBI BioProject accession: PRJNA384311 and SRA accessions: SRR3030980, SRR2182495, SRR2182497, SRR2182499, and SRR2182501. Source data are provided with this paper.

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

## Acknowledgements

We thank Dr. Guanghui Wang, Ping Xiang, and Zhe Tang for their invaluable laboratory assistance. We thank Professor Junfeng Liu from China Agricultural University for generously providing us with the prokaryotic expression vector pRSFDuet-1. We also thank lab technicians Qiong Zhang and Xiaona Zhou from the State Key Laboratory for Crop Stress Resistance and High-Efficiency Production for their assistance with the MS/MS analysis. This work was supported by grants from the National Key R&D Program of China (2022YFD1400102) for H.L., the National Natural Science Foundation of China (no. 32170200 for H.L. and 32300171 for C.F.), the China Postdoctoral Science Foundation (2023M742873) for K.X., and the China National Postdoctoral Program for Innovative Talents (BX20230296) for C.F.

## Author contributions

H.L. C.F., K.X. and Y.D. conceived and designed the experiments; C.F., K.X., Y.D., J.Z., X.X., Q.X., Y.Z. and R.Z. performed the experiments; H.L., W.H., Q.W., C.J., X.W., Z.K. and J-R.X. contributed materials/analysis tools and analyzed the data; H.L., C.F., K.X. and Y.D. wrote the paper. C.F., K.X., Y.D. and J.Z. contributed equally to this paper.

## Competing interests

The authors declare no competing interests.
