## [Peer Review File · Nature Communications]

Unveiling the A-to-I mRNA editing machinery and its regulation and evolution in fungiREVIEWER COMMENTS

Reviewer #1 (Remarks to the Author):

In this study, the authors identified a protein complex regulation mRNA editing in the filamentous fungus *Fusarium graminearum*. This complex is made of two proteins known to be implicated in tRNA editing (Tad2 and Tad3) associated with a newly identified protein (Ame1) which seems to act as an activator of mRNA editing during sexual development. Ame1 expression is restricted to sexual stage whereas Tad2 and Tad3 expression and function are regulated through alternative promoter, translation start site as well as post-translational modification, the resulting alternative isoforms having different capacities in heterologous interactions. Interesting, this new editing complex which is sufficient to induce editing in yeast, *E. coli* and human cells. seems to be specific of some Sordariomycetes. This is a very interesting paper and the amount of work presented is spectacular. Yet, some minor aspects of the presented experiments need to be clarified. My main comment concerns the potential role of Ame1 which is presented as an activator of the editing without further comments. Maybe the discussion could provide more hypothesis for its mode of action..

Minor comments:

- 1) Line 101 : It is was the data suggest. There is not proof of that. This sentence needs to be edited.
- 2) Figure 1e. It is unclear from where this data comes. No result are presented, no reference are given.
- 3) Fig1f. Edited and not edited transcripts are not presented. We can guess that it is coming from another study which not cited.
- 4) Line 116. There is no proof for that. The fact that it is binding mRNA does not prove the activity. This sentence should be edited.
- 5) It is not clear was the MEME motif is supposed to described in the figure 1 as it is not discussed in the text.
- 6) Figure 1g. Overexpression is convincing but the comparison between sex and hyp in WT would also bring info and could be added in this panel
- 7) Figure1h. I don't know what I am supposed to look at it this panel and what is supporting the fact that vegetative growth is normal and that the sexual development is poorly affected.
- 8) Figure 2d. No WT is shown as control for the morphology of ascospores.
- 9) How the T2Ssil-P and T2del-P has been validated? What is the impact of these mutations on the L transcript expression.
- 10) Why the deletion of the L transcript is not tested? It is essential?
- 11) TAD3 S and L transcripts encodes the same protein. What is the impact of the deletion of the short transcript on TAD3 expression?. What would be the deletion of the long one?
- 12) Line 194. "from" is missing
- 13) I am surprised that the author did not confirm the conclusion from the fig 2g through N-terminal tagging of the protein. An alternative possibility is that the Ssil-P transcript is less stable.
- 14) The identification of the start codon of the S-transcript would have been more convincing using a N-terminal tagging strategy. Yet this information, is not crucial for the understanding of the paper.
- 15) Lines 433 to 442 are duplicated.

Reviewer #2 (Remarks to the Author):

Chanjing and colleagues have identified the A-to-I mediating enzyme/complex in fungi for the first time. The authors show that in fungi, as in bacteria, TadA (FgTad2 and FgTad3), an enzyme previously thought to be dedicated to tRNA editing also performs mRNA editing. Additionally, they have identified AME1 as a novel regulator of mRNA editing. This is very interesting and suggests that other proteins with importance for RNA editing may exist in other organisms. Thus, the work will be of significance to the field of RNA editing, microbiology, and mycology. Notably, I am no expert in fungi biology, so it is important that a reviewer from this field evaluates

the manuscript in terms of the biology and phenotypes observed.

This work is novel and important. However, the authors should correct some of the text, especially acknowledging the work on TadA-dependent mRNA editing in bacteria and add some experiments for validation. In general, after correcting the manuscript it should interest the broad readership of Nature Communications.

Comments and required experiments:

>Lines 57-58: The statement "the bacterial TadA enzyme only requires the presence of the anticodon-stem-loop structure of the cognate tRNAArgACG for activity in vitro" is not true.

>TadA requires the anticodon-stem-loop structure AND a UACG motif. This was shown both in the original work identifying the enzyme (Wolf et al 2002) and in the work that first described the occurrence of TadA-dependent mRNA editing in bacteria (Bar Yaacov et al 2017).

>The current work is novel, but IT IS NOT THE FIRST TIME TadA WAS SHOWN TO EDIT mRNA. The authors are requested to clearly mention that TadA-dependent mRNA editing activity was reported in bacteria (Bar-Yaacov et al 2017).

>Lines 101-102: What supports the statement "Like their yeast orthologs, FgTad2 serves as the catalytic subunit, while FgTad3 plays a structural role"? There is no reference to a figure or results. The authors should explain their statement or add evidence or references for this statement.

>Figure 2b – please clarify how many biological replicates were used. Did each time point was conducted once?

>Line 164 – please give a short explanation of the rationale for using the mat1-1-1 and mat1-2-1 strains. It is a deletion mutant of what? I could not find the explanation in the manuscript.

>Figure 2e – how do the authors explain the reduction in editing in the mat1-1-1 and 1-2-1 strains? This is why it is important to explain in the text the rationale for using these strains.

>Importantly, the T2Sdel-p removes some amino acids of the long protein while the T2Ssil-p strain does not. The strongest effect is on the T2Sdel-p strain, while the T2Ssil-p strain shows a similar editing phenotype as the mat1-1-1/1-2-1 strains. The authors are requested to clearly explain, in light of this result, why they still think it is the shorter version of FgTad2 that is more important for mRNA editing.

>Why is the mat1-1-1 strain not found in the FgTad3 figure in fig. 2e?

>Why did the authors group T2Sdel-p and T2Ssil-p strains for the statistical analysis in fig. 2e bottom panel? It is clear to me that there is NO difference between and the T2Ssil-p strain mat1-1-1/2 strains. Why is not compared to the WT strain?

>Figure 2d-e – Currently, the reduction in editing and the phenotype in fig. 2d are better explained by the deletion of some amino acids in the long version of Tad2. The authors should clearly state this possibility.

>Figure 2f – in line 153 the authors state that the transcription start site of the short version of Tad2 is "located between 280-310 bp downstream of the start codon of the L-transcript". So the long version should be at around 100 amino acid longer than the short version. The authors should clearly state in the text the expected size of the two versions when they refer to figure 2f. This will help in understanding the figure.

>IMPORTANT: with regard to Figure 2 as a whole - I am convinced that there are two isoforms of FgTad2. However, and most importantly, I want to see validation of the RNA-seq experiments using Sanger sequencing. The authors are requested to choose several edited sites (5-10 sites) and show a panel of Sanger sequencing of DNA and RNA (cDNA) samples of all strains in Figure 2e.

>The authors report the identification of AME1 and report that the Δ ame1 strain does not have

editing. Did the authors perform their analysis with a WT control? How many biological replicates were used? Even if they did not find editing in the Δ ame1 strain, the authors should add the results in a figure containing the mRNA editing sites and levels of editing of the Δ ame1 strain side-by-side with the result from a WT control.

>Figure 3f – please explain in the legend what the numbers on the x-axis represent.

>Please add a supplementary figure showing, using Sanger sequencing the increase in editing in a few (3-5) sites for the strains shown in Figure 3f (or at least for AME-oe). For example WT Hyp vs AME-oe HYP.

>Figure 6a – the authors describe a self-editing site in AME and FgTad3. This self-editing site is mentioned first in the text in line 442. The authors should explain: what is this self-editing site? How did the authors identify it? Does it occur endogenously in *Fusarium graminearum*? Why did they focus on this site and not others in the yeast, hek293 and bacterial RNA editing analysis?

>How did the authors exclude DNA mutation in the population sequenced? Did they sequence corresponding DNA and RNA samples in Figure 6? As AME1 is a novel protein, its activity on the DNA should be excluded. At least, the authors are requested to sample some sites by DNA and RNA Sanger sequencing to validate their claim of mRNA editing.

>Figure 6b – The color of AG in the legend does not match the color in the graph itself. The authors are requested to change it so they will fit.

Point-by-point responses to the reviewers' comments

Reviewer #1 (Remarks to the Author):

In this study, the authors identified a protein complex regulation mRNA editing in the filamentous fungus *Fusarium graminearum*. This complex is made of two proteins known to be implicated in tRNA editing (Tad2 and Tad3) associated with a newly identified protein (Ame1) which seems to act an activator of mRNA editing during sexual development. Ame1 expression is restricted to sexual stage whereas Tad2 and Tad3 expression and function are regulated through alternative promoter, translation start site as well as post-translational modification, the resulting alternative isoforms having different capacities in heterologous interactions. Interesting, this new editing complex which is sufficient to induce editing in yeast, *E. coli* and human cells. seems to be specific of some Sordariomycetes. This is a very interesting paper and the amount of work presented is spectacular. Yet, some minor aspects of the presented experiments need to be clarified.

Response: We deeply appreciate your acknowledgment of our study and your valuable comments and suggestions.

My main comment concerns the potential role of Ame1 which is presented as an activator of the editing without further comments. Maybe be the discussion could provide more hypothesis for its mode of action.

Response: Ame1 is a protein of unknown function that contains a DUF726 domain belonging to the alpha/beta hydrolase superfamily. In our study, we demonstrated that Ame1 interacts with the N-terminal domain of FgTad3, and this interaction is essential for mRNA editing. Despite Ame1 belonging to the alpha/beta hydrolase superfamily, its function in mRNA editing appears independent of the putative catalytic activity of alpha/beta hydrolases. Therefore, we hypothesized that this interaction may lead to a conformational change in the FgTad2-FgTad3 complex, potentially diminishing the specificity for recognizing substrate RNA structures. We have included this discussion in the revised manuscript.

Minor comments:

1) Line 101 : It is was the data suggest. There is not proof of that. This sentence needs to be edited.

Response: We found both FgTad2 and FgTad3 contain the typical CDA domain with conserved residues required for activities in their catalytic core, but the conserved E residue essential for catalysis was replaced by a V residue in FgTad3 as its yeast ortholog (Fig. 1a). These observations suggest that FgTad2 serves as the catalytic subunit and FgTad3 plays a structural role. To avoid misunderstanding, we have revised this sentence as “*FgTad2 and FgTad3 act as a heterodimer for adenosine deamination in F. graminearum, akin to their yeast counterparts.*” in the revised manuscript.

2) Figure 1e. It is unclear from where this data comes. No results are presented, no references are given.

Response: The mRNA editing sites used for sequence logo analysis were obtained from our previous study (Feng et al., 2022, mBio, doi:10.1128/mbio.01872-22). The sequences of eight inosine-modified tRNAs in *F. graminearum* were obtained from the GtRNAdb database. We added descriptions and references in the revised manuscript. The relevant sentences in the Results section were also revised as: *“In addition to being located within a similar stem-loop structure, WebLogo analysis of all the identified A-to-I mRNA editing sites¹² and A³⁴ sites of inosine-modified tRNAs in F. graminearum showed similar base preferences (Fig. 1e)”*

3) Fig1f. Edited and not edited transcripts are not presented. We can guess that it is coming from another study which was not cited.

Response: Edited and unedited transcripts (genes) were categorized based on the identified mRNA editing sites in our previous study (Feng et al., 2022, mBio, doi:10.1128/mbio.01872-22). Descriptions have been added in the revised manuscript. *“Based on the known A-to-I mRNA editing sites identified in our previous study¹², all genes expressed during sexual reproduction were classified into edited and unedited groups using a binary approach. Additionally, we further divided the edited genes into high-edited and low-edited groups based on the editing intensity (calculated by summing the editing levels of all editing sites) of each gene. We compared the normalized read density between the binary groups.”*

4) Line 116. There is no proof for that. The fact that it is binding mRNA does not prove the activity. This sentence should be edited.

Response: Sorry for the confusion. We were not using mRNA binding capacity (footprint read density) to demonstrate RNA editing activity. The editing activity was determined based on the categorization of high-edited or low-edited genes with varying editing intensity (by summing the editing levels of all editing sites). To clarify our statement, the revised sentence reads as follows: *“These results indicate that FgTad2-FgTad3 binds to mRNA of edited genes, with a preference for highly edited genes, suggesting their involvement in mRNA editing in F. graminearum.”*

5) It is not clear what the MEME motif is supposed to describe in the figure 1 as it is not discussed in the text.

Response: We assume that the MEME motif you referred to is the WebLogo in Figure 1e, which illustrates the base preference of tRNA and mRNA editing. Figure 1e was cited in the Results section in our previous manuscript. To clarify this point, we revised the relevant sentence as follows: *“In addition to being located within a similar stem-loop structure, WebLogo analysis of all the identified A-to-I mRNA editing sites¹² and A³⁴ sites of inosine-modified tRNAs in F. graminearum showed similar base preferences (Fig. 1e)”*.

6) Figure 1g. Overexpression is convincing but the comparison between sex and hyp in WT would also bring info and could be added in this panel

Response: Thank you for your suggestion. Because of the comparison of different categories, it is difficult to put them in the same panel. The expression levels of *FgTAD2* and *FgTAD3* in sexual and hyphal samples of the wild type were included as Supplementary Fig. 1 in the revised manuscript.

7) Figure 1h. I don't know what I am supposed to look at in this panel and what is supporting the fact that vegetative growth is normal and that the sexual development is poorly affected.

Response: In the revised manuscript, we added arrows to indicate the observed defects (transparent perithecia) on this panel. To enhance clarity in our descriptions, we also revised the relevant sentence in Results section as follows: "*Both transformants exhibited normal colony growth and morphology. Although the perithecia produced by these transformants appeared transparent, they were still able to generate regular asci and ascospores internally (Fig. 1h).*"

8) Figure 2d. No WT is shown as control for the morphology of ascospores.

Response: The morphology of ascospores derived from self-fertilization of the WT is shown in Figure 2d. We assume that by "WT" you are referring to the control for the outcrosses involving the *mat1-1-1* deletion mutant labeled with H1-GFP. *Fusarium graminearum* is homothallic and tends to undergo self-mating. This outcross experiment aims to assess the female fertility of self-sterile mutants. We employed the self-sterile *mat1-1-1* deletion mutant instead of the WT as the male strain in the outcrosses to avoid the formation of normal perithecia through self-fertilization of the male strain. Since the *mat1-1-1* deletion mutant is also self-sterile, the observation of a 1:1 segregation of 8 ascospores with and without GFP signals in an ascus clearly indicates that the self-sterile mutant is fertile as a female; otherwise, no ascospore formation would occur. Therefore, the control is deemed unnecessary. Nevertheless, we have included the outcross of the *mat1-1-1* deletion mutant (male) and WT (female) as a control in the revised manuscript.

9) How the T2S^{sil-P} and T2S^{del-P} has been validated? What is the impact of these mutations on the L transcript expression?

Response: Good question. Desired mutations in the T2S^{sil-P} and T2S^{del-P} mutants were confirmed through PCR and Sanger DNA sequencing. Quantitative PCR was also utilized to verify the absence of any unintended integration of allelic fragments in the mutants. Given that the *FgTAD2* S-transcript promoter resides within the coding region of the L-transcript, the *FgTAD2* L-transcript in the T2S^{del-P} mutant deleted the S-transcript promoter, inevitably expresses a truncated protein missing amino acid residues 29-88 before the CDA domain. Throughout our investigation, we were particularly concerned that alterations in the L-transcript protein could impact the phenotype. Consequently, we engineered the T2S^{sil-P} mutant, which incorporates

multiple synonymous mutations introduced in the S-transcript's promoter region with the aim of suppressing S-transcript transcription without influencing the L-transcript protein. The T2S^{sil-P} mutant displayed similar defects to the T2S^{del-P} mutant, suggesting that the deletion of amino acid residues 29-88 of the L-transcript protein did not contribute additionally to the observed phenotype. Furthermore, Normal phenotypes observed in the *FgTAD2*^{M1,2,3L} mutant, which produces a truncated protein by the L-transcript lacking all amino acid residues before the M4 codon, including 29-88aa (Fig. 2h), indicates that the defective phenotype observed in the T2S^{del-P} mutant is not attributed to alterations in the L transcript protein.

During our manuscript review, we examined the RNA-seq data utilized for identifying A-to-I mRNA editing sites in perithecia of the T2S^{sil-P} and T2S^{del-P} mutants. We observed that the expression level of the L-transcript in the T2S^{sil-P} mutant showed no noticeable changes compared to the wild type, suggesting that the synonymous mutations had minimal effects on the stability of the L-transcript. Hence, the T2S^{sil-P} mutant is suitable for investigating the roles of the *FgTAD2* L-transcript in sexual reproduction and mRNA editing. Surprisingly, our RNA-seq data revealed an increased expression of the L-transcript in the T2S^{del-P} mutant relative to the wild type, indicating that deletion of the promoter of the S-transcript enhances L-transcript expression. The similar phenotype observed between the T2S^{del-P} and T2S^{sil-P} mutants indicates that increased L-transcript expression does not compensate for the absence of the S-transcript, highlighting the critical roles of the S-transcript. Since the underlying mechanism for the enhanced expression of the L-transcript remains to be elucidated, and the T2S^{sil-P} mutant is sufficient for clarifying the importance of the S-transcript, we have excluded information about the T2S^{del-P} mutant in the revised manuscript. Additionally, we have included Supplementary Fig. 2 to depict L-transcript expression in the T2S^{sil-P} mutant and WT in the revised manuscript.

10) Why the deletion of the L transcript is not tested? It is essential?

11) TAD3 S and L transcripts encodes the same protein. What is the impact of the deletion of the short transcript on TAD3 expression? What would be the deletion of the long one?

Response: Still good questions. Because these two comments are related, we responded to them together. As both *TAD2* and *TAD3* are essential genes expressing only the L-transcript in hyphae, the L-transcript of *TAD2* and *TAD3* appears to be vital for viability. However, we have successfully generated deletion mutants for the L-transcript promoters of *TAD2* and *TAD3* and found the resulting mutants displayed normal phenotypes in growth. This is because the deletion leads to the activation of a truncated transcript isoform utilizing downstream transcription start sites in hyphae. Moreover, our results demonstrated that deletion of the S-transcript promoter results in increased expression of the L-transcript in hyphae for both *TAD2* and *TAD3*. These results suggest that the promoter regions of both L- and S-transcripts are regulated by chromatin. Since these complex outcomes extend beyond the scope of the current manuscript and their

omission does not affect our conclusions, we are currently preparing another manuscript focused on elucidating the regulatory evolution of *TAD2* and *TAD3*.

12) Line 194. “from” is missing

15) Lines 433 to 442 are duplicated.

Response: All corrected. We appreciate your attention to detail.

13) I am surprised that the author did not confirm the conclusion from the fig 2g through N-terminal tagging of the protein. An alternative possibility is that the T2S^{sil-P} transcript is less stable.

14) The identification of the start codon of the S-transcript would have been more convincing using a N-terminal tagging strategy. Yet this information, is not crucial for the understanding of the paper.

Response: These two comments are related, and we responded to them together. Firstly, our RNA-seq data revealed that the expression level of the L-transcript in the T2S^{sil-P} mutant showed no obvious changes compared to the wild type, suggesting that the synonymous mutations did not visibly impact the stability of the L-transcript. Secondly, the *FgTAD2* L-transcript generates two protein isoforms through alternative translational initiation utilizing the M1 and M4 codons as start codons, respectively. The transcriptional initiation site of the *FgTAD2* S-transcript is located within the coding region of the L-transcript before the M4 codon. The start codon of the *FgTAD2* S-transcript corresponds to the M4 codon of the L-transcript. Both transcripts produce M4-FgTad2. When applying N-terminal tagging to M1-FgTad2, both transcripts yield M4-FgTad2 without the tag. Conversely, when utilizing N-terminal tagging for M4-FgTad2, both transcripts produce M4-FgTad2 with the tag. Consequently, the N-terminal tagging strategy is unable to differentiate the origin of M4-FgTad2 between the two transcripts. Moreover, considering the scanning mechanism involved in translation initiation on eukaryotic mRNAs, adding an N-terminal tag could potentially influence start codon selection. Therefore, we opted for a C-terminal tagging strategy in our study.

Reviewer #2 (Remarks to the Author):

Chanjing and colleagues have identified the A-to-I mediating enzyme/complex in fungi for the first time. The authors show that in fungi, as in bacteria, TadA (FgTad2 and FgTad3), an enzyme previously thought to be dedicated to tRNA editing also performs mRNA editing. Additionally, they have identified AME1 as a novel regulator of mRNA editing. This is very interesting and suggests that other proteins with importance for RNA editing may exist in other organisms. Thus, the work will be of significance to the field of RNA editing, microbiology, and mycology. Notably, I am no expert in fungi biology, so it is important that a reviewer from this field evaluates the manuscript in terms of the biology and phenotypes observed. This work is novel and important. However, the authors should correct some of the text, especially acknowledging the work on TadA-dependent mRNA editing in bacteria and add some experiments for validation. In general, after correcting the manuscript it should interest the broad readership of Nature Communications.

Response: We deeply appreciate your acknowledgment of our study and your valuable comments and suggestions.

Comments and required experiments:

>Lines 57-58: The statement “the bacterial TadA enzyme only requires the presence of the anticodon-stem-loop structure of the cognate tRNA^{Arg}_{ACG} for activity in vitro” is not true. TadA requires the anticodon-stem-loop structure AND a UACG motif. This was shown both in the original work identifying the enzyme (Wolf et al 2002) and in the work that first described the occurrence of TadA-dependent mRNA editing in bacteria (Bar Yaacov et al 2017). The current work is novel, but IT IS NOT THE FIRST TIME TadA WAS SHOWN TO EDIT mRNA. The authors are requested to clearly mention that TadA-dependent mRNA editing activity was reported in bacteria (Bar-Yaacov et al 2017).

Response: We acknowledge your point that both the anticodon-stem-loop structure and the UACG motif are essential for tRNA editing by TadA. However, our statement, *"While the bacterial TadA enzyme only requires the presence of the anticodon-stem-loop structure of the cognate tRNA^{Arg}_{ACG} for activity in vitro, the eukaryotic tRNA-specific heterodimeric deaminase Tad2-Tad3 requires the complete tertiary structure of cognate tRNAs to perform its deamination reaction and is not known to have mRNA editing capacity."*, is intended to provide a structural perspective on the differences between bacterial TadA and eukaryotic Tad2-Tad3. We do not disregard the importance of other factors in editing. Furthermore, our manuscript only reported the mRNA editing capacity of eukaryotic Tad2-Tad3. As you pointed out, previous studies have reported TadA-dependent mRNA editing activity in bacteria (Bar-Yaacov et al., 2017), as well as widespread off-target A-to-I editing in cellular mRNAs using TadA-derived adenine base editors (Rees et al., 2019). In contrast to bacterial TadA, eukaryotic Tad2-Tad3 is known to rely strictly on the complete tertiary structure of cognate tRNAs for editing, making it challenging to envision its ability to edit mRNA. The reference you

suggested has been included in the revised manuscript. The relevant sentences were revised as follows: “*TadA* has been associated with a small number of naturally occurred A-to-I mRNA editing sites in bacteria^{26, 27}. Overexpression of *TadA*-derived adenine base editors can induce widespread A-to-I editing in cellular mRNAs²⁸. However, the eukaryotic tRNA-specific heterodimeric deaminase *Tad2-Tad3* typically requires the complete tertiary structure of cognate tRNAs for deamination and is not known for mRNA editing capability^{29, 30}.”

>Lines 101-102: What supports the statement “Like their yeast orthologs, FgTad2 serves as the catalytic subunit, while FgTad3 plays a structural role”? There is no reference to a figure or results. The authors should explain their statement or add evidence or references for this statement.

Response: We found both FgTad2 and FgTad3 contain the typical CDA domain with conserved residues required for activities in their catalytic core, but the conserved E residue essential for catalysis was replaced by a V residue in FgTad3 as its yeast ortholog (Fig. 1a). These observations suggest that FgTad2 serves as the catalytic subunit and FgTad3 plays a structural role. In the revised manuscript, we have revised the relevant sentences as follows: “*FgTad2* and *FgTad3* act as a heterodimer for adenosine deamination in *F. graminearum*, akin to their yeast counterparts.” in the revised manuscript.

>Figure 2b – please clarify how many biological replicates were used. Did each time point was conducted once?

Response: The RNA-seq data utilized for transcript expression analysis of *FgTAD2* and *FgTAD3* have only one biological replicate for each time point. However, given that RNA-seq data are primarily employed to illustrate transcript expression patterns rather than to pinpoint significantly differentially expressed genes/transcripts, the absence of multiple biological replications does not impact our conclusions. The consistent expression trends of *FgTAD2* and *FgTAD3* across the 1- to 8-dpf samples indicate the reliability of RNA-seq data. In the revised manuscript, details regarding these RNA-Seq datasets have been documented in Supplementary Data-5.

>Line 164 – please give a short explanation of the rationale for using the *mat1-1-1* and *mat1-2-1* strains. It is a deletion mutant of what? I could not find the explanation in the manuscript.

>Figure 2e – how do the authors explain the reduction in editing in the *mat1-1-1* and *1-2-1* strains? This is why it is important to explain in the text the rationale for using these strains.

>Why did the authors group T2S^{del-P} and T2S^{sil-P} strains for the statistical analysis in fig. 2e bottom panel? It is clear to me that there is NO difference between the T2S^{sil-P} strain and *mat1-1-1/2* strains. Why is not compared to the WT strain?

>Importantly, the T2S^{del-P} removes some amino acids of the long protein while the

T2S^{sil-P} strain does not. The strongest effect is on the T2S^{del-P} strain, while the T2S^{sil-P} strain shows a similar editing phenotype as the *mat1-1-1/1-2-1* strains. The authors are requested to clearly explain, in light of this result, why they still think it is the shorter version of FgTad2 that is more important for mRNA editing.

Response: These four comments are related, and we responded to them together. The *mat1-1-1* and *mat1-2-1* strains are deletion mutants of the mating type genes *MAT1-1-1* and *MAT1-2-1*, well-known for their crucial role in mating across most filamentous ascomycetes. In *F. graminearum*, the *mat1-1-1* and *mat1-2-1* deletion mutants displayed impaired sexual reproduction during self-fertilization, resulting in the formation of small perithecia without ascogenous hyphal development (Zheng et al., 2013, doi:10.1371/journal.pone.0066980). Previous studies have shown that fungal mRNA editing activities predominantly occurs in asci. Therefore, mutants lacking asci, such as the *mat1-1-1* and *mat1-2-1* mutants, may show reduced mRNA editing compared to the WT. The absence of asci in both T2S^{del-P} and T2S^{sil-P} mutants clearly underscores the critical role of the S-transcript in sexual development but hinders our assessment of its true involvement in mRNA editing in asci. To further explore the roles of the S-transcript in mRNA editing, we utilized the *mat1-1-1* and *mat1-2-1* mutants as controls for developmental issues. Given that both T2S^{del-P} and T2S^{sil-P} mutants related the suppression of the S-transcript, we grouped them together for statistical analysis in comparison with the two *mat* control strains.

As synonymous mutations in promoter regions may only partially suppress S-transcript transcription rather than completely block it, the reduction of RNA editing in the T2S^{sil-P} mutant was not as pronounced as in the T2S^{del-P} mutant. For editing level comparison, we only considered the shared editing sites among the *mat*, T2S^{del-P} and T2S^{sil-P} mutants. The median editing levels of shared editing sites in the T2S^{sil-P} mutant are not statistically significant compared to those in the *mat* mutants. In the revised manuscript, we have excluded information about the T2S^{del-P} mutant (Please refer to our response regarding comment #9 from Reviewer #1). The results indicated that the median editing levels of the T2S^{sil-P} mutant were significantly lower than those of the *mat1-1-1* and *mat1-2-1* mutants. These results highlight the role of the *FgTAD2* S-transcript in mRNA editing. However, it is crucial to note that since the S-transcript is primarily expressed in ascus tissues, comparing with the *mat* mutants can only demonstrate that the S-transcript indeed have an impact on RNA editing, but may not fully reveal its complete effect.

To improve the clarity of our results, we have revised relevant paragraphs in the revised manuscript. The reference for the *mat1-1-1* and *mat1-2-1* mutants has been cited.

>Why is the *mat1-1-1* strain not found in the FgTad3 figure in fig. 2e?

Response: Since the T3S^{del-P} mutant displayed defects solely in ascospore formation without affecting ascus formation, it is suitable to compare its mRNA editing profiles with the wild type. Conversely, the *mat* mutants lacking asci are not suitable for comparison due to their distinct developmental defects.

>Figure 2d-e – Currently, the reduction in editing and the phenotype in fig. 2d are better explained by the deletion of some amino acids in the long version of Tad2. The authors should clearly state this possibility.

Response: Several lines of evidence indicate that the deficiencies in mRNA editing and sexual development in the T2S^{del-P} mutant are not linked to the deletion of specific amino acids in the long version of Tad2. In the T2S^{del-P} mutant, the *FgTAD2* L-transcript expresses a truncated protein lacking amino acid residues 29-88 before the CDA domain. However, the amino acids encoded by the *FgTAD2* L-transcript in the T2S^{sil-P} mutant remain unchanged. Our RNA-seq data also revealed that the expression level of the L-transcript in the T2S^{sil-P} mutant had no noticeable alterations compared to the wild type. Despite the deletion of amino acid residues 29-88 from the L-transcript protein, the T2S^{del-P} mutant exhibited similar defects to the T2S^{sil-P} mutant, suggesting that this deletion did not contribute obviously to the observed phenotype. Furthermore, the lack of phenotypic changes in the *FgTAD2*^{M1,2,3L} mutant, which generates a truncated protein via the L-transcript omitting all amino acid residues before the M4 codon, including 29-88aa, further supports that the defective phenotype seen in the T2S^{del-P} mutant is not a result of alterations in the L transcript protein.

However, since the T2S^{sil-P} mutant is sufficient for clarifying the importance of the S-transcript, we have excluded information about the T2S^{del-P} mutant in the revised manuscript. Please refer to our response regarding comment #9 from Reviewer #1.

>Figure 2f – in line 153 the authors state that the transcription start site of the short version of Tad2 is “located between 280-310 bp downstream of the start codon of the L-transcript”. So the long version should be at around 100 amino acid longer than the short version. The authors should clearly state in the text the expected size of the two versions when they refer to figure 2f. This will help in understanding the figure.

Response: Thank you for your suggestion. It is true that M1-FgTad2 has a 112 amino acid N-terminal extension compared to M4-FgTad2. As advised, we have included the expected sizes of the two protein variants in the revised manuscript.

>IMPORTANT: with regard to Figure 2 as a whole - I am convinced that there are two isoforms of FgTad2. However, and most importantly, I want to see validation of the RNA-seq experiments using Sanger sequencing. The authors are requested to choose several edited sites (5-10 sites) and show a panel of Sanger sequencing of DNA and RNA (cDNA) samples of all strains in Figure 2e.

Response: It is important to note that the strains used for DNA-seq and RNA-seq, as well as the reference genome sequences employed for RNA editing identification, are all derived from the same PH-1 laboratory stock. The reference genome sequences of PH-1 are highly accurate, with minimal repeat sequences (Lu et al, 2022, doi: 10.1111/nph.18164). To enable comparison, an equivalent number of deduplicated mapped reads were randomly sampled for each of the compared samples. These measures ensure that our RNA editing analysis is as robust as possible. The

identification of only a few non-A-G variant sites (please refer to Supplementary Table 1) in individual samples indicates the accuracy of our identification results.

As per your request, we have included Supplementary Fig. 3 to present the validation of six randomly selected editing sites in the T2S^{sil-P} mutant through Sanger sequencing of DNA and RNA samples. The results demonstrate the reliability of our RNA-seq analysis. With the profound reduction in both the number and editing levels of detected A-to-I mRNA editing sites in the T3S^{del-P} mutant, Sanger sequencing validation of a few sites is deemed unnecessary. Nevertheless, we appreciate your meticulous approach to scientific research.

>The authors report the identification of AME1 and report that the $\Delta ame1$ strain does not have editing. Did the authors perform their analysis with a WT control? How many biological replicates were used? Even if they did not find editing in the $\Delta ame1$ strain, the authors should add the results in a figure containing the mRNA editing sites and levels of editing of the $\Delta ame1$ strain side-by-side with the result from a WT control.

Response: Yes, due to the defective early stage of sexual development in the $\Delta ame1$ mutant, we conducted RNA-seq analysis of A-to-I mRNA editing in the wild type, $\Delta ssc23$, and $\Delta ame1$ mutants during 60-hour sexual cultures with two biological replicates for each. The $\Delta ssc23$ mutant, which exhibits more severe defects than $\Delta ame1$, was utilized as a control for developmental issues. An average of 152 A-to-I mRNA editing sites were identified in the wild type and 26 in the $ssc23$ mutant. Despite its milder defects, no reliable A-to-I mRNA editing sites were detected in the $\Delta ame1$ mutant. Additionally, RNA-seq analysis of 6-dpf perithecia of the $\Delta ame1$ mutant was performed to confirm the absence of A-to-I mRNA editing. Since no A-to-I mRNA editing sites were identified in the $\Delta ame1$ mutant, we were unable to present the mRNA editing sites and levels of editing for this mutant. In the revised manuscript, Supplementary Fig. 7 has been included to showcase A-to-I mRNA editing sites detected in the wild type and $\Delta ssc23$ mutant but not in the $\Delta ame1$ mutant. Detailed information regarding identified RNA variants and RNA-Seq datasets for these strains has been outlined in Supplementary Table 1 and Supplementary Data-5, respectively.

>Figure 3f – please explain in the legend what the numbers on the x-axis represent.

Response: The numbers on the x-axis correspond to strain numbers. Figure 3f has been revised in the revised manuscript.

>Please add a supplementary figure showing, using Sanger sequencing the increase in editing in a few (3-5) sites for the strains shown in Figure 3f (or at least for AME-oe). For example WT Hyp vs AME-oe HYP.

Response: As per your request, we have included Supplementary Fig. 8 to show the validation of 6 edited sites through Sanger sequencing of DNA and RNA samples. It is important to note that A-to-I mRNA editing is specific to the sexual stage, and we only detected one A-to-I site in hyphal samples of the WT. Therefore, the large number of A-to-I mRNA editing sites identified in *AME1*-overexpressing transformants is

adequate to demonstrate the role of *AME1*. In the revised manuscript, detailed information regarding identified RNA variants and the RNA-Seq dataset for the WT hyphal samples has been provided in Supplementary Table 1 and Supplementary Data-5, respectively.

>Figure 6a – the authors describe a self-editing site in AME and FgTad3. This self-editing site is mentioned first in the text in line 442. The authors should explain: what is this self-editing site? How did the authors identify it? Does it occur endogenously in *Fusarium graminearum*? Why did they focus on this site and not others in the yeast, hek293 and bacterial RNA editing analysis?

Response: The self-editing sites refer to the A-to-I mRNA editing sites that naturally occur in *AME1* and *FgTAD3* in *F. graminearum*. When examining the A-to-I mRNA editing activities of the FgTad2-FgTad3-Ame1 complex in heterologous systems, as we are unaware in advance of which mRNAs in the heterologous systems serve as substrates, we utilized the self-editing sites in *AME1* and *FgTAD3* as targets to evaluate the editing activity. This approach eliminates the need for the transformation of additional substrate targets to assess the editing activity. To improve the clarity of our description, we have revised relevant sentences in the revised manuscript.

>How did the authors exclude DNA mutation in the population sequenced? Did they sequence corresponding DNA and RNA samples in Figure 6? As *AME1* is a novel protein, its activity on the DNA should be excluded. At least, the authors are requested to sample some sites by DNA and RNA Sanger sequencing to validate their claim of mRNA editing.

Response: Yes, for the identification of A-to-I mRNA editing, we utilized matched DNA-seq and RNA-seq data from the same samples. Any DNA mutations have been excluded through filtering. Therefore, the identified A-to-I mRNA editing by our RNA-seq data is dependable. Our main aim was to analysis the RNA editing profiles using RNA-seq analysis, rather than focusing on individual sites. Since Sanger sequencing of PCR products is not as precise in evaluating RNA editing levels as RNA-seq, Sanger sequencing validation of a few sites does not significantly strengthen our conclusion and is therefore considered unnecessary. Nevertheless, we value your meticulous approach to scientific research. In the revised manuscript, Supplementary Fig. 10 has been included to showcase A-to-I mRNA editing sites detected in RNA-Seq data but not DNA-seq data of the yeast INvSC1 strain and *E. coli* BL21 strain expressing the FgTad2-FgTad3-Ame1 complex.

>Figure 6b – The color of AG in the legend does not match the color in the graph itself. The authors are requested to change it so they will fit.

Response: Corrected. We appreciate your attention to detail.

REVIEWERS' COMMENTS

Reviewer #1 (Remarks to the Author):

no further questions
thank you

Reviewer #2 (Remarks to the Author):

A lot of work has been done in the current and revised manuscript. The authors have answered all of my comments.

The only suggestion I have is regarding my comment:

"Figure 2b – please clarify how many biological replicates were used. Did each time point was conducted once?"

The authors only have one biological replication for the RNA-seq analysis. So I think they should clearly write it in the text or material and methods and perhaps explain their rationale as they did in their response. Personally, I expect to see experiments in duplicates, at least, if not triplicates. However, I do not ask the authors to do this if they do not wish.

I recommend accepting the paper for publication.

Point-by-point responses to the reviewers' comments

Reviewer #1 (Remarks to the Author):

no further questions

thank you

Reviewer #2 (Remarks to the Author):

A lot of work has been done in the current and revised manuscript. The authors have answered all of my comments.

The only suggestion I have is regarding my comment:

"Figure 2b – please clarify how many biological replicates were used. Did each time point was conducted once?"

The authors only have one biological replication for the RNA-seq analysis. So I think they should clearly write it in the text or material and methods and perhaps explain their rationale as they did in their response. Personally, I expect to see experiments in duplicates, at least, if not triplicates. However, I do not ask the authors to do this if they do not wish.

I recommend accepting the paper for publication.

Response: We have added the relevant descriptions in a new included section titled "Statistics & Reproducibility" in the methods section. Thank you for your suggestions.